# Disruption of tRNA biogenesis enhances proteostatic resilience, improves later-life health, and promotes longevity

Yasir Malik[1]ⓘ, Yavuz Kulaberoglu[2]ⓘ, Shajahan Anver[2]ⓘ, Sara Javidnia[2], Gillian Borland[3], Rene Rivera[1], Stephen Cranwell[2], Danel Medelbekova[2], Tatiana Svermova[2], Jackie Thomson[3], Susan Broughton[4], Tobias von der Haar[1], Colin Selman[3]*, Jennifer M. A. Tulletⓘ[1]*, Nazif Alicⓘ[2]*

1 School of Biosciences, University of Kent, Canterbury, United Kingdom, 2 Institute of Healthy Ageing, Research Department of Genetics Evolution and Environment, University College London, London, United Kingdom, 3 School of Molecular Biosciences, College of Medical, Veterinary and Life Sciences, University of Glasgow, Glasgow, United Kingdom, 4 Division of Biomedical and Life Sciences, Faculty of Health and Medicine, Lancaster University, Lancaster, United Kingdom

ⓘ These authors contributed equally to this work.
* colin.selman@glasgow.ac.uk (CS); j.m.a.tullet@kent.ac.uk (JMAT); n.alic@ucl.ac.uk (NA)

**Data Availability Statement:** Raw sequencing data have been submitted to Gene Expression Omnibus projects: GSE232719, GSE232720, GSE232721,

## Abstract

tRNAs are evolutionarily ancient molecular decoders essential for protein translation. In eukaryotes, tRNAs and other short, noncoding RNAs are transcribed by RNA polymerase (Pol) III, an enzyme that promotes ageing in yeast, worms, and flies. Here, we show that a partial reduction in Pol III activity specifically disrupts tRNA levels. This effect is conserved across worms, flies, and mice, where computational models indicate that it impacts mRNA decoding. In all 3 species, reduced Pol III activity increases proteostatic resilience. In worms, it activates the unfolded protein response (UPR) and direct disruption of tRNA metabolism is sufficient to recapitulate this. In flies, decreasing Pol III's transcriptional initiation on tRNA genes by a loss-of-function in the TFIIIC transcription factor robustly extends lifespan, improves proteostatic resilience and recapitulates the broad-spectrum benefits to late-life health seen following partial Pol III inhibition. We provide evidence that a partial reduction in Pol III activity impacts translation, quantitatively or qualitatively, in both worms and flies, indicating a potential mode of action. Our work demonstrates a conserved and previously unappreciated role of tRNAs in animal ageing.

## Introduction

The rate of ageing differs widely between species, with ageing virtually absent in some while others have lifespans measured in days [1,2]. Despite these differences, the cellular and organismal processes that modulate the rate of ageing are surprisingly conserved [3–5]. Understanding these evolutionarily conserved processes driving loss of physiological function and promoting pathology and disease during ageing presents an opportunity to maintain human health at older ages. The urgent requirement for new, safe, and effective points of intervention

GSE232723 and GSE232724. Additional data are available as Supplementary Data.

**Funding:** This work was funded by the Biotechnology and Biological Sciences Research Council (BBSRC) grant BB/S014330/1 to CS, BB/S014365/1 to JMAT and BB/S014357/1 to NA, and partially by BBSRC grant BB/W013525/1 to NA and Leverhulme Trust grant RPG-2022-181 to JMAT. The funders had no role in study design, data collection and analysis, decision to publish, or preparation of the manuscript.

**Competing interests:** The authors have declared that no competing interests exist.

**Abbreviations:** ER, endoplasmic reticulum; FDR, false discovery rate; LM, linear mode; PCA, principal component analysis; RQC, Ribosome-associated Quality Control; SYA, sugar/yeast/agar; TF, transcription factor; TPM, transcripts per million; UPR, unfolded protein response.

for human ageing is underscored by the global rise in the proportion of elderly people within society and the inexorable increase in the burden of age-related diseases, many of which remain untreatable [6,7].

The rate of ageing can be slowed by changes to gene expression, e.g., by remodelling cellular transcriptional programmes [5]. In the eukaryotic nucleus, these programmes are set by the activities of 3 DNA-dependent RNA polymerases that share the labour of transcription [8,9]. The largest of these, RNA polymerase (Pol) III is dedicated to the generation of abundant, short, noncoding RNAs that perform various fundamental cellular functions [8,9]. These include tRNAs, adaptor molecules required for translation; the 5S rRNA, a component of the ribosome; several RNAs involved in RNA processing, such as the U6, an RNA component of the spliceosome; as well as RNAs of other functions such as the 7SL, an RNA component of the signal recognition particle, and 7SK, an RNA involved in regulation of elongation by Pol II [8–11].

Despite the fundamental importance of Pol III and the genes it transcribes, research prioritising protein-coding genes had left the role of Pol III in animal physiology relatively unexplored. Our previous work showed that Pol III activity drives ageing: partial inhibition of Pol III extends lifespan in yeast, worms, and flies [12,13]. In humans, genetic data also indicate that tissue-specific expression of Pol III subunits inversely correlates with longevity [14], while recent evidence in mice indicates Pol III may contribute to certain age-related pathologies, such as osteoporosis [15].

In this study, to better understand how Pol III affects ageing, we set out to delineate the Pol III-transcribed genes responsible for its effects on lifespan and late-life health. We show that partial inhibition of Pol III does not impact equally on the expression of all Pol III-transcribed genes, rather it predominantly disrupts tRNA levels and this regulatory pattern is conserved between worms, flies, and mice. We find that both Pol III inhibition and disruptions to tRNA metabolism trigger increased proteostatic resilience. In worms and flies, Pol III inhibition has measurable consequences on protein translation. Crucially, using fruit fly genetics, we find that inhibition of Pol III initiation at tRNA loci underpins a robust extension of lifespan and confers broad benefits to late-life health. Our findings highlight the critical and conserved role played by tRNAs in modulating the rate of animal ageing.

## Results

### Partial loss of function in Pol III specifically impacts tRNA expression in *C. elegans*

Pol III transcribes 100 s of short, noncoding RNAs involved in several fundamental cellular processes [9,10]. To better understand which are involved in ageing, we sought to define the Pol III-transcribed RNAs whose levels are altered by the longevity-promoting, partial loss-of-function in Pol III. We compared expression profiles of adult *Caenorhabditis elegans* that were treated with RNAi against the gene encoding the largest subunit of Pol III, *rpc-1*, from the L4 stage to that of controls exposed to the vector alone; this *rpc-1* RNAi treatment results in partial knockdown of the cognate mRNA and promotes longevity [12]. We employed a modified RNA-Seq protocol, similar to published protocols [16], including size-selection and demethylation steps to enrich for Pol III-transcribed RNAs, which are about 70 to 300 bases long and often heavily modified. The reads obtained were used to quantify transcript levels using Salmon, which is able to discriminate RNAs similar in sequence and estimate their levels [17]. Differential expression was determined with DESeq2 [18], for all transcripts shorter than 300 bases at a false discovery rate (FDR) of 10%. We obtained consistent signals for all main,

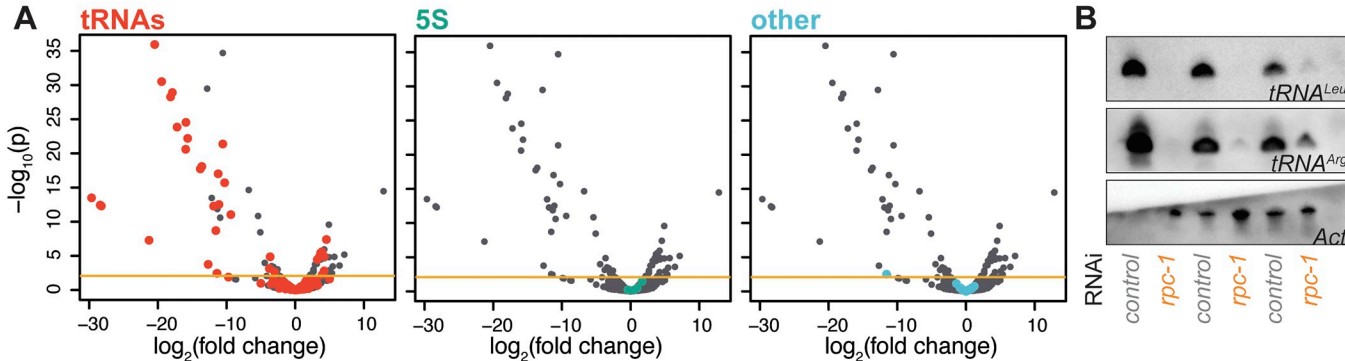

**Fig 1. Partial inhibition of Pol III in worms alters tRNA expression specifically.** (A) Volcano plots of differential expression of short RNAs (<300 b) between worms treated with *rpc-1* RNAi and vector controls, with different classes of transcripts coloured in different panels. "Other" refers to snoRNAs identified in a previous study (see main text). Horizontal line indicates the p value threshold for 10% FDR. (**B**) Northern blots of RNA from worms treated with the vector control or *rpc-1* RNAi. *Actin* band from 1 wild-type sample was accidentally lost from the blot. Data underlying this figure can be found in S1 Data. Uncropped images of blots are presented in S1 Fig.

expected RNAs, including tRNAs, 5S rRNA, and a number of snoRNAs known to be transcribed by Pol III in worms [19].

We found that not all Pol III-transcribed genes were equally sensitive to the partial loss of function in the polymerase, detecting widespread changes in the amounts of different tRNAs in the worms treated with *rpc-1* RNAi but no substantial effects on 5S, 7SL, and others (Fig 1A, full analysis results are given in S1 Data). Indeed, we detected 35 tRNAs as differentially expressed. We independently confirmed the substantial and reproducible reduction in the levels of a *tRNA^Leu* and a *tRNA^Arg* by northern blotting (Figs 1B and S1). Interestingly, although the majority of tRNAs were repressed, 9 (26%) were induced. As these changes are a result of Pol III inhibition, the most parsimonious explanation is that a reduction in some tRNAs triggers a compensatory increase in others. Overall, our analysis indicated that the longevity phenotype of *rpc-1* knockdown is associated with a disturbance in expression of tRNAs specifically, but not of other Pol III-transcribed genes.

## Pol III inhibition and impairments in tRNA metabolism can activate UPR

Reduced activity of Pol III in fruit flies results in resistance to tunicamycin [12], an agent that causes proteostatic stress in the endoplasmic reticulum (ER). To test whether this is conserved in *C. elegans*, we quantified the ability of control and *rpc-1* RNAi-treated animals to withstand tunicamycin exposure. Similarly to flies, we observed that worms are significantly more tunicamycin resistant upon partial inhibition of Pol III (Fig 2A and S1 Table). Additionally, we found that *rpc-1* RNAi also enhanced *C. elegans* resistance to a pulse of heat stress (Fig 2B and S1 Table), a marker of general proteostatic tolerance in worms [20]. These resistance phenotypes appeared specific for stresses that disturb proteostasis, as *rpc-1* RNAi treatment did not induce tolerance to juglone, a xenobiotic thought to cause oxidative stress [21] (S2A Fig and S2 Table). Together, these results suggest that a reduction in Pol III activity causes a conserved increase in proteostatic resistance.

Such increased proteostatic resistance may be caused by the activation of the unfolded protein response (UPR) [22]. To establish whether *rpc-1* RNAi triggers UPR activation, we examined *C. elegans* carrying copies of the well-established *hsp-4*::*GFP* reporter [23] using quantitative fluorescence microscopy. HSP-4 is the *C. elegans* homologue of BiP, the ER resident chaperone whose expression increases in response to ER stress [24,25]. Indeed, we found

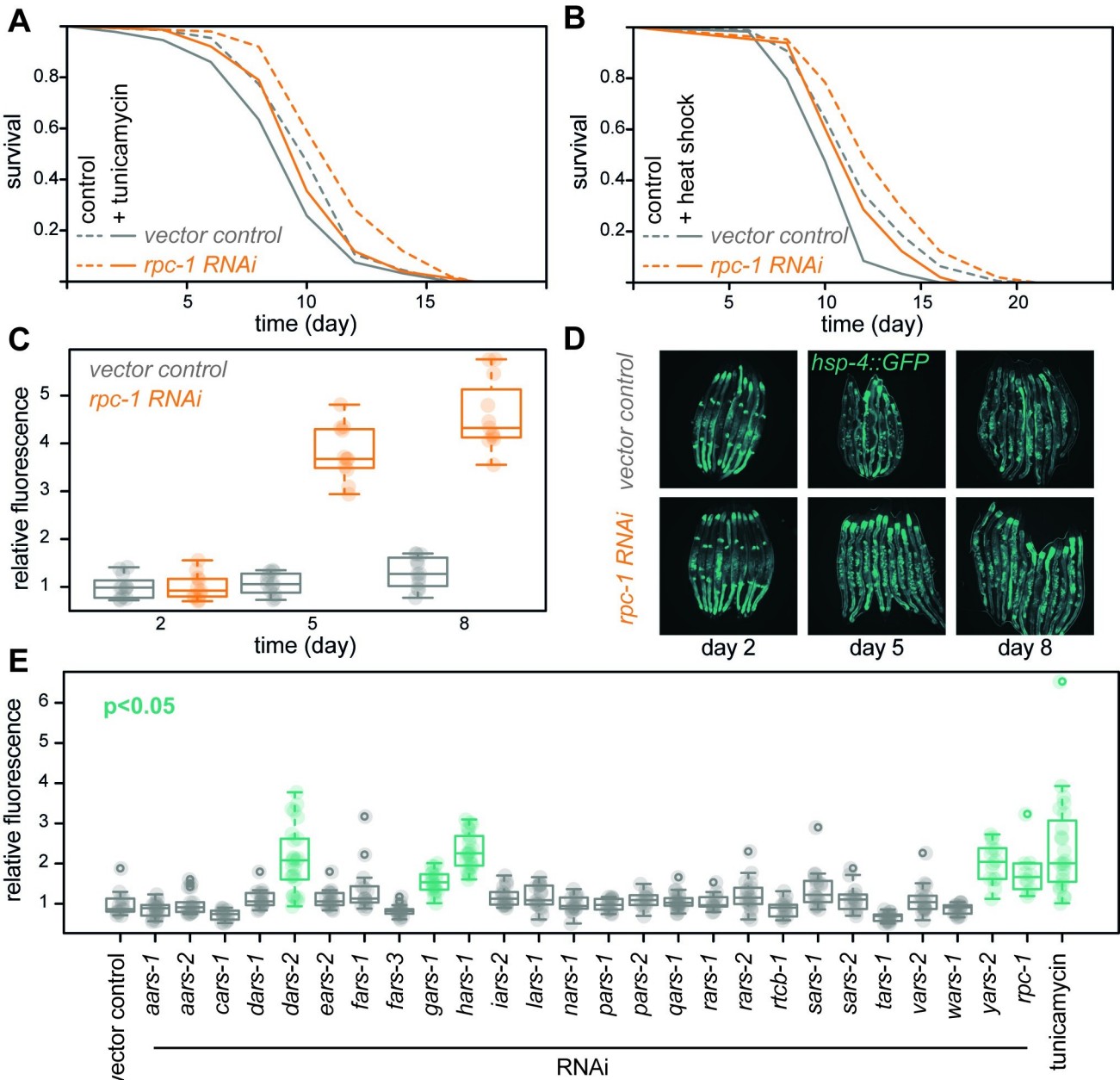

**Fig 2. Pol III inhibition and interference in tRNA metabolism both trigger UPR in worms.** (**A**) Tunicamycin and (**B**) heat shock survival of worms treated with *rpc-1* RNAi or the vector control. Demography and statistical analyses of data presented in A, B and additional repeats are provided in S1 Table. (**C**) Expression of the *hsp-4-4::GFP* reporter in *rpc-1* RNAi- or vector control-treated worms ($n = 11$ *rpc-1* RNAi/12 control; effect of RNAi $p < 10^{-4}$, effect of age $p < 10^{-4}$, RNAi by age interaction $p < 10^{-4}$, *LM*). Fluorescence intensities are relative to control on day 2. (**D**) Representative images. (**E**) Expression of the *hsp-4::GFP* reporter (day 5 of adulthood) upon RNAi against genes encoding aminoacyl tRNA synthases, as well as *rpc-1* RNAi and tunicamycin treatment ($n = 10–25$; *ANOVA* $p < 10^{-4}$; $p$ value from *Dunnet's test* to vector control is indicated). Boxplots show quantiles with individual data points overlayed. Data underlying this figure can be found in S1 Data.

that *rpc-1* knockdown increased the expression of *hsp-4::GFP* as worms aged, indicating that the UPR is also induced in these animals (Fig 2C and 2D).

Given that Pol III inhibition is specifically impacting tRNA expression, we examined whether the UPR could also be activated by directly interfering with tRNA metabolism. Using RNAi, we individually knocked down 25 aminoacyl tRNA synthases, enzymes responsible for

charging tRNAs with their cognate amino acids, and quantified *hsp-4*::*GFP* expression. Interestingly, we found that individually knocking down several aminoacyl tRNA synthases was sufficient to activate the expression of *hsp-4*::*GFP* to the levels similar to those achieved by *rpc-1* RNAi or tunicamycin exposure (Fig 2E). One of these was *dars-2*, an aspartate-tRNA synthase, and indeed, a $tRNA^{Asp}$ (WormBase transcript ID: F28F8.t3) was the second most repressed tRNA in *rpc-1*-knockdown worms (S1 Data). Hence, altering the availability of specific, charged tRNAs is sufficient to alter *C. elegans* physiology, namely UPR activation, consistent with alterations caused by Pol III inhibition.

The UPR consists of 3 branches each activating a specific transcription factor (TF): XBP-1, ATF-4, and ATF-6 [22]. Mutations in each TF was sufficient to reduce the expression of *hsp-4*::*GFP* and abolish the ability of *rpc-1* RNAi to induce it (S2B–S2D Fig). Given that each UPR branch has been associated with health and longevity [26–34], we then tested whether reducing the activity of each individual TF was sufficient to reduce the lifespan extension caused by Pol III inhibition. As previously reported, RNAi against *rpc-1* was able to extend lifespan in wild-type worms [12] and this effect was maintained in worms mutant for *xbp-1* or *atf-4* or when *rpc-1* and *atf-6* RNAi were co-administered (S2E–S2G Fig and S2 Table), suggesting that no UPR branch alone is required for *rpc-1* RNAi mediated longevity. Similarly, individually knocking down the aminoacyl tRNA synthases, whose loss-of-function was able to significantly induce *hsp-4*::*GFP* (Fig 2E), was not sufficient to induce longevity (S2H Fig). In contrast, broad, pharmacological inhibition of these enzymes in yeast and *C. elegans* [35], or a mutation in a valyl aminoacyl tRNA synthetase (*vars-2*) in *C. elegans* [36], can all extend lifespan, possibly because they cause a more substantial loss of cellular tRNA-charging ability. Hence, while we could not ascribe the longevity of *rpc-1* knockdown to a specific branch of UPR in *C. elegans*, our data indicate that partial inhibition of Pol III increased tolerance to proteostatic stress, likely mediated by a disruption to tRNA metabolism.

## Pol III loss-of-function impacts tRNA expression and activates UPR in flies

Pol III activity limits lifespan in both worms and flies, showing evolutionary conservation [12]. We wanted to examine whether the differential effect on tRNA expression by a partial loss of function in Pol III can also be observed in flies. We compared expression profiles between wild-type and long-lived female flies carrying a heterozygous mutation (a P-element insertion) in one of the Pol III-specific subunits, *Polr3D* (previously named *dC53*, see ref. [37]) at 1 week of age. This mutation represents a substantial loss of function: homozygous flies are not viable while heterozygous females have a significant reduction in *Polr3D* mRNA [12]. We focused on the adult fly midgut, since Pol III activity in this organ drives fly ageing [12].

Similarly to *C. elegans*, we found that not all Pol III-transcribed genes were equally sensitive to the partial loss of function in the polymerase, detecting extensive changes in the amounts of tRNAs in the mutant guts, with a limited or no effects on 5S RNA and others (Fig 3A, "Others" identified in ref. [38]; full analysis given in S1 Data). Again, as observed in *C. elegans*, we found some tRNAs were repressed and others induced; surprisingly, however, in $Polr3D^{EY/+}$ flies the majority were induced (Fig 3A). Additionally, the magnitude of change was substantially less in flies than in worms (compare x axes in Figs 1A and 3A). The results of northern blot analyses of a $tRNA^{His}$ and a $tRNA^{Gln}$ expression were consistent with the RNA-Seq findings (Fig 3B with the results of Linear Model (*LM*) analysis presented in the caption, S3A–S3D Fig).

tRNAs are produced by Pol III as precursors that are subsequently processed to mature forms [39,40] and the level of these pre-tRNAs is often a good indicator of in vivo Pol III activity. The surprising finding of increases in abundance of several mature tRNAs in $Polr3D^{EY/+}$

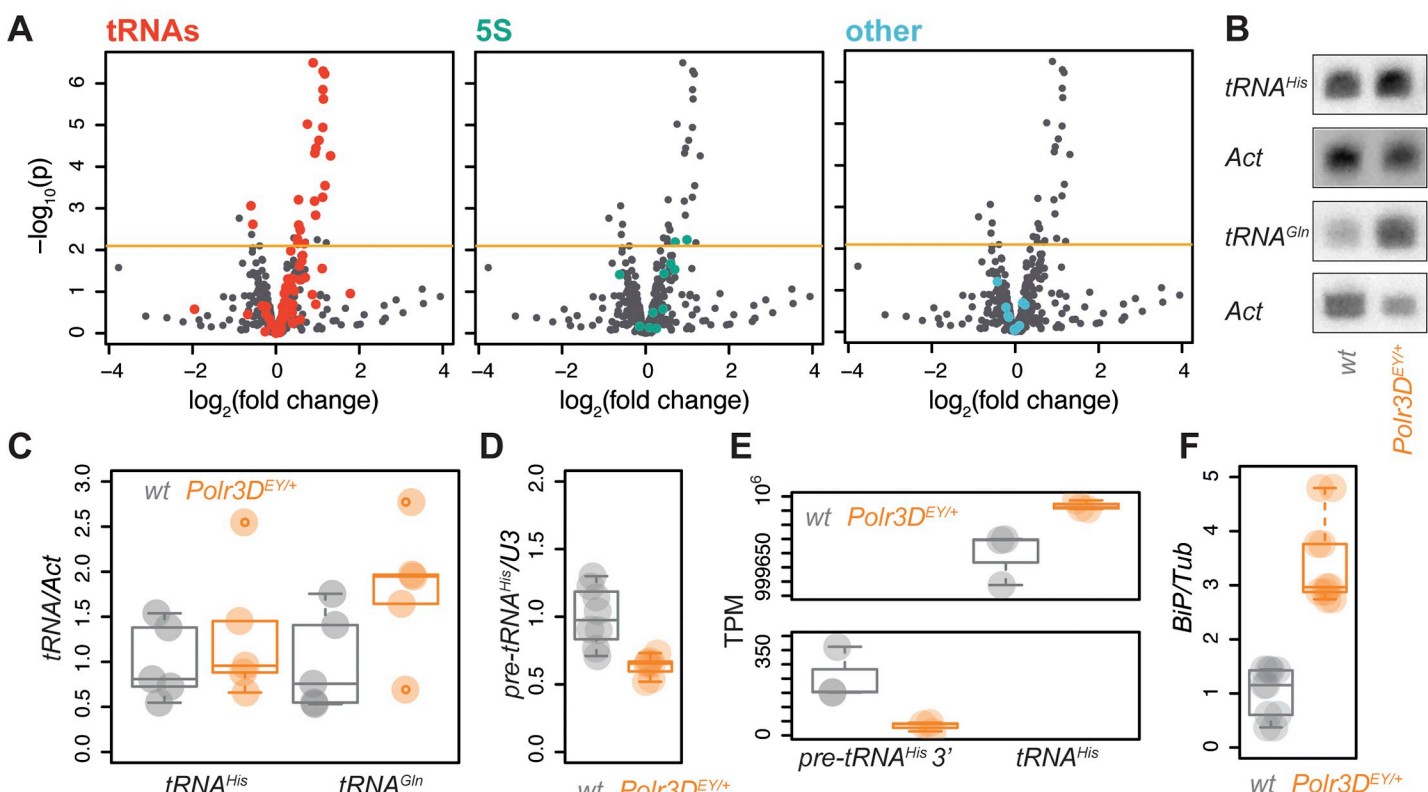

**Fig 3. Partial inhibition of Pol III in flies alters tRNA expression and induces *BiP*.** (**A**) Volcano plots of differential expression of short RNAs (<300 b) in midguts of female *Polr3D$^{EY/+}$* flies relative to wild type, with different classes of transcripts coloured in different panels. "Other" refers to *U6* and several snoRNAs identified in a previous study (see main text). Horizontal line indicates the *p* value threshold for 10% FDR. (**B**) Northern blots of RNA from dissected guts, representative images. (**C**) Northern blots, quantification of relative tRNA expression (*n* = 5, Polr3DEY/+ versus wild type *p* = 0.035 *one-tailed t test* on least-squares means within *LM*; *LM* summary: effect of genotype *p* = 0.07, no significant effect of tRNA type or the genotype by type interaction). (**D**) Relative expression of *pre-tRNA$^{His}$* assessed by qPCR (*n* = 7–8, *p* = 1.5 × 10$^{-3}$ *t test*). (**E**) TPM estimated in the RNA-Seq data for 3′ processed region of *pre-tRNA$^{His}$* and mature *tRNA$^{His}$* (*n* = 3, *Polr3D$^{EY/+}$* versus wild type for pre-tRNA and tRNA separately *p* < 0.05 post hoc *t tests* on least-squares means within *LM*; *LM* summary: effect of genotype *p* > 0.05, effect of tRNA sequence *p* < 10$^{-4}$, genotype by sequence interaction *p* = 2 × 10$^{-3}$). (**F**) Relative expression of *BiP* mRNA in whole bodies of female *Polr3D$^{EY/+}$* flies (*n* = 10, *p* < 10$^{-4}$ *t test*). Boxplots show quantiles with individual data points overlayed. Data underlying this figure can be found in S1 Data. Uncropped images of blots are presented in S3 Fig.

flies prompted us to quantify pre-tRNAs in the fly gut. Using qPCR, we found the expression levels of *pre-tRNA$^{His}$* reduced (Fig 3C), which contrasted with the increase in the mature *tRNA$^{His}$* levels observed in RNA-Seq and northern blot (S1 Data and Fig 3B). This differential effect on *pre-tRNA$^{His}$* and mature *tRNA$^{His}$* was confirmed when we reanalysed the RNA-Seq data to estimate specifically the occurrence of sequences corresponding to the 3′ end of *pre-tRNA$^{His}$*, which is absent from the mature form, and the mature *tRNA$^{His}$*. As expected, transcripts per million (TPM) estimated for the precursor were reduced in *Polr3D$^{EY/+}$* flies, while those corresponding to the mature tRNA were increased (Fig 3D; note that *pre-tRNA$^{His}$* TPMs are several orders of magnitude lower than for the mature tRNA). This indicates that compensation for partial loss of Pol III activity is occurring in the fly gut, for example by reduced tRNA degradation. This compensation is likely more extensive than seen in worms since the mutant flies had impaired Pol III function from development onwards. The differential impact on the expression of different tRNA observed by RNA-Seq may not be simply due to this feedback, as *Polr3D* loss of function had no significant effect on the expression levels of 2 other pre-tRNAs, *pre-tRNA$^{Ile}$* and *pre-tRNA$^{Leu}$* (S3E and S3F Fig). Hence, both in worms and flies, loss-of-function in Pol III results in a complex disturbance of tRNA expression.

Similarly to worms, partial Pol III inhibition increased the fly's resistance to the proteotoxic stress caused by tunicamycin [12]. We wanted to examine if it also causes UPR activation. As a readout of UPR activation, we examined the transcript levels of the ER chaperone Hsc70-3/ BiP (orthologue of the worm HSP-4). Expression of *BiP* was significantly increased in *Polr3-D$^{EY/+}$* females (Fig 3E), consistent with increased tunicamycin resistance. Overall, our analyses revealed that the longevity due to a partial loss of function in Pol III is associated with changes in tRNA abundance and UPR activation in flies, as in worms.

## Impairing initiation on tRNA genes is sufficient to extend *Drosophila* lifespan

Our RNA-Seq analyses indicated complex alteration occurring specifically in tRNA levels upon partial loss of function in Pol III in both worms and flies. Pol III initiates transcription from 3 types of promoters. Each promoter type is used to recruit the polymerase to a specific class of genes and requires a particular combination of TFs [9,41] (Fig 4A): TFIIIA is solely

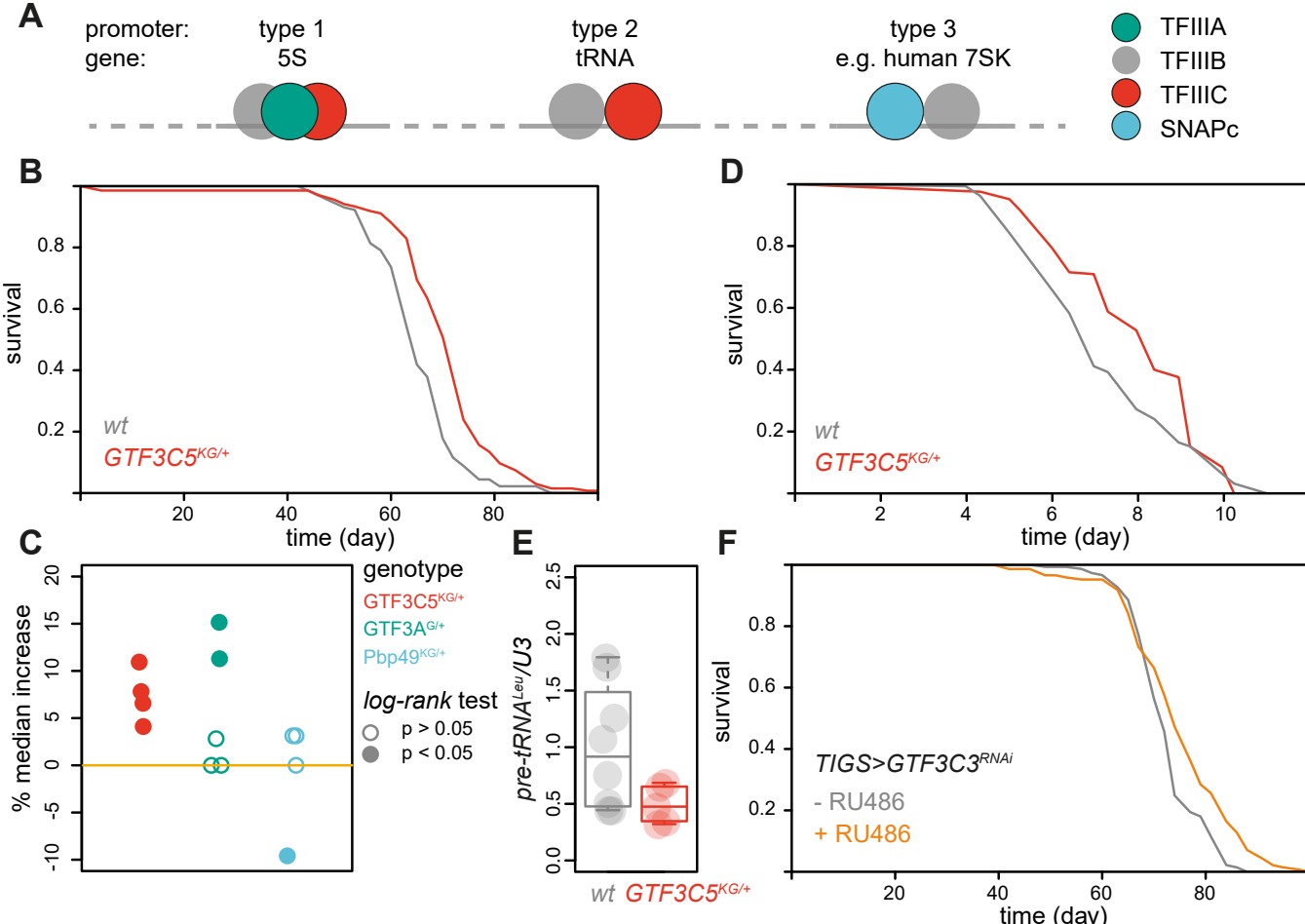

**Fig 4. TFIIIC activity promotes ageing in the fruit fly.** (**A**) Schematic of promoter types and TFs used by Pol III to initiate transcription. (**B**) Lifespans of *GTF3C5$^{KG/+}$* and wild-type female flies ($n$ = 134/4 and 124/7 dead/censored flies respectively, $p$ = 2.3 × 10$^{-13}$ *log-rank test*). (**C**) Changes in median lifespan relative to wild-type observed for the indicated genotypes in several experimental trials (see S4 Fig for details; $p$ values from *log-rank test* are indicated). (**D**) Tunicamycin resistance of *GTF3C5$^{KG/+}$* and wild-type female flies ($n$ = 165/0 and 158/0 dead/censored flies, respectively, $p$ = 1.3 × 10$^{-8}$ *log-rank test*). (**E**) Relative expression of *pre-tRNA$^{Leu}$* ($n$ = 5–8, $p$ = 0.04 *t test*). (**F**) Lifespans of *TIGS>GTF3C3$^{RNAi}$* in the presence or absence of the RU486 inducer ($n$ = 144/2 and 147/2 dead/censored flies respectively, $p$ = 2.2 × 10$^{-4}$ *log-rank test*). Data underlying this figure can be found in S1 Data.

required for the expression of 5S (type 1 promoter), TFIIIC for expression of 5S and tRNAs (types 1 and 2), while SNAPc is required for expression of other RNAs such as 7SK [9,41] (type 3). To directly examine which Pol III targets contribute to ageing, we decide to manipulate TFs that are required for initiation by the polymerase, as this allowed us to restrict the classes of genes whose expression is affected.

We employed the fly model as TFIIIA, TFIIIC, and SNAPc could be distinguished in the genome of this organism. Using Flybase, we identified the *CG9609* (henceforth *GTF3A*) as the *Drosophila* orthologue of the human *GTF3A* gene encoding TFIIIA, *l(2)37Cd* (henceforth *GTF3C5*) as encoding a subunit of TFIIIC and *Pbp49* encoding a subunit of SNAPc. We obtained P element insertions into these genes, namely $GTF3A^{G2025}$, $GTF3C5^{KG04199}$, and $Pbp49^{KG02425}$, and backcrossed them into an outbred, wild-type background. We compared the lifespans of heterozygous mutant and wild-type females, since Pol III inhibition has a pronounced effect on longevity in that sex [12]. TFIIIC heterozygous mutant flies were consistently long lived in all 4 independent, sequential experimental trials performed (Figs 4B, 4C and S4). On the other hand, TFIIIA heterozygous mutants were long-lived in 2 out of 5 trials (Figs 4C and S4) indicating TFIIIA may have a conditional or restricted effect on ageing. SNAPc mutants were not observed to be long-lived in any experimental trial (Figs 4C and S4). This partial loss-of-function in TFIIIC was also sufficient to cause tunicamycin resistance (Fig 4D) as previously observed upon Pol III inhibition [12]. This phenotype appeared specific to the UPR since TFIIIC heterozygous mutant flies were not resistant to the oxidative stress-inducing agent, paraquat [42] (S4J Fig).

We also found similarities between the effects of $GTF3C5^{KG04199/+}$ and $Polr3D^{EY/+}$ on pre-tRNA expression in the gut: both resulted in a decrease in expression of at least 1 pre-tRNA tested without affecting all pre-tRNAs equally. Specifically, the mutation in the gene encoding the TFIIIC subunit resulted in a decrease in $pre\text{-}tRNA^{Leu}$ levels (Fig 4E), without significant effects on $pre\text{-}tRNA^{His}$ and $pre\text{-}tRNA^{Ile}$ (S5A and S5B Fig). Hence, as expected, TFIIIC heterozygous mutant flies show evidence of impaired initiation from a type 2 promoter.

Partial Pol III inhibition restricted to the adult female fly gut is sufficient for longevity [12]. To extend and further validate our TF analysis, we knocked down *GTF3A*, *CG8950* (henceforth *GTF3C3*) encoding another subunit of TFIIIC or the *Pbp95* subunit of SNAPc specifically in the adult female gut, using available and validated RNAi lines and the inducible, GeneSwitch driver, TIGS. The RNAi constructs were induced from day 2 of adulthood by feeding the RU486 inducer; the inducer had no significant effect on the lifespan of the driver-alone control (S5C Fig). Similar to the mutant phenotypes described above, we observed an extension of lifespan upon adult gut-restricted knockdown of the TFIIIC subunit that was not present when TFIIIA or a SNAPc subunit were knocked down (Figs 4F, S5D and S5E); *GTF3A* and *Pbp95* knockdowns actually shortened lifespan. Hence, using independent approaches and genetic reagents we established TFIIIC activity as driving ageing in the fruit fly. Together with the transcriptomic analysis presented above, the consistent effect of TFIIIC on longevity strongly implicates tRNAs as Pol III targets limiting lifespan, with the expression of 5S potentially having a more restricted, condition- or tissue-specific role.

## Mutation in a TFIIIC subunit recapitulates broad health benefits of Pol III inhibition

Akin to humans, flies display a range of age-related pathologies and functional impairments. With age, the intestinal stem cells in their gut hyperproliferate and mis-differentiate and the ability of their gut to act as a barrier becomes compromised, with these phenotypes being more prominent in females [43–45]. Our previous work showed that the inhibition of Pol III is

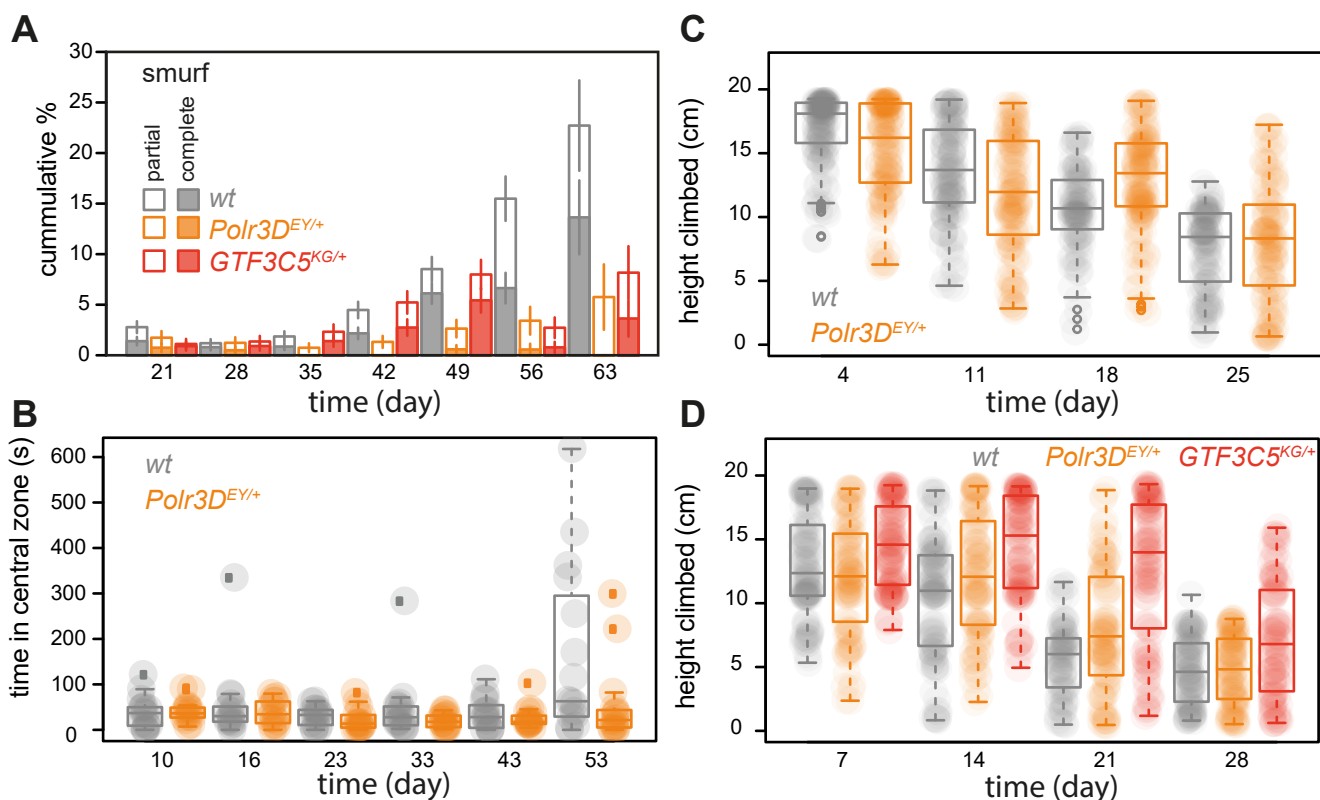

**Fig 5. Pol III and TFIIIC limit fly health in late life.** (**A**) Cumulative proportion of partial and full smurfs (flies with impaired gut barrier function) in cohorts of $Polr3D^{EY/+}$, $GTF3C5^{KG/+}$ and wild-type females ($n = 52$–788, effect of genotype: mutants *versus* wild-type $p = 0.11$ and $GTF3C5^{KG/+}$ *versus* $Polr3D^{EY/+}$ $p = 0.88$, age $p < 2 \times 10^{-16}$, age by genotype interaction: mutants *versus* wild-type $p = 9.8 \times 10^{-4}$ and $GTF3C5^{KG/+}$ *versus* $Polr3D^{EY/+}$ $p = 0.42$, *ordinal logistic regression*). Mean and standard error of the mean shown. (**B**) Time spent in the central zone of the arena during exploratory walking by $Polr3D^{EY/+}$ and wild-type females ($n = 14$–16, effect of genotype $p = 0.01$, age $p = 2.7 \times 10^{-3}$, age by genotype interaction $p = 9 \times 10^{-3}$, *LM*). (**C**) Hight climbed by $Polr3D^{EY/+}$ and wild-type females in a climbing (negative geotaxis) assay ($n = 99$–126, effect of genotype $p = 4.7 \times 10^{-6}$, age $p < 2 \times 10^{-16}$, age by genotype interaction $p = 2.7 \times 10^{-6}$, *LM*). (**D**) Hight climbed by $Polr3D^{EY/+}$, $GTF3C5^{KG/+}$ and wild-type females in a climbing assay ($n = 95$–108, effect of genotype: mutants *versus* wild-type $p = 0.09$ and $GTF3C5^{KG/+}$ *versus* $Polr3D^{EY/+}$ $p = 1.7 \times 10^{-3}$, age $p < 2 \times 10^{-16}$, age by genotype interaction: mutants *versus* wild-type $p = 0.046$ and $GTF3C5^{KG/+}$ *versus* $Polr3D^{EY/+}$ $p = 0.32$; *LM*). Boxplots show quantiles with individual data points overlayed. Data underlying this figure can be found in S1 Data.

able to delay this age-related decline in gut health [12]. We next examined whether a mutation in a TFIIIC subunit can recapitulate this, by looking at the increase in the number of flies with compromised gut barrier function (those that were unable to restrict the presence of a blue food dye to their gut—the "smurf" phenotype [44,46]). Similarly to humans, flies in our out-bred, healthy background display substantial heterogeneity in ageing phenotypes (e.g., ref. [47]). Hence, to insure robust statistical analyses, we employed appropriate regression models to detect effects on the rate of ageing within the population as a significant age-by-genotype interaction. We found that both a heterozygous mutant in the $GTF3C5$ TFIIIC subunit and in the $Polr3D$ Pol III subunit were able to significantly delay the age-related increase in the number of flies with a compromised gut barrier function with age (Fig 5A, statistical analysis using *ordinal logistic regression* and a priori contrasts is given in S6A Fig). We found no evidence that the magnitude of their effects on this phenotype was different between $GTF3C5^{KG04199}$ and $Polr3D^{EY/+}$ (S6A Fig), indicating both mutants were similarly able to delay age-associated impairments in gut health.

As flies age, the performance of their neuronal and muscular systems also deteriorates. This deterioration can be observed as age-related changes in their spontaneous, exploratory walking

behaviour, as well as in their ability to climb a vertical surface in an induced, negative geotaxis response [47–49]. Comparing the performance of $Polr3D^{EY/+}$ and wild-type females, we found that $Polr3D^{EY/+}$ delayed the age-related increase in the time spent in the centre of the arena during exploratory walking (Fig 5B, statistical analysis using a *LM* is given in S6B Fig), with no effect on other age-related changes in exploratory behaviour (S6B–S6E Fig). Similarly, $Polr3$-$D^{EY/+}$ delayed age-related loss in climbing ability (Fig 5C). Hence, in addition to its effect on gut health, the reduction in Pol III activity improves the performance of the neuromuscular system with age.

Lastly, we used a climbing assay to examine whether the improvements to neuromuscular performance in old age can be recapitulated by impairing TFIIIC function. Both $GTF3C5^{KG04199}$ and $Polr3D^{EY/+}$ females showed a significant delay in the age-related decline in climbing ability, with no significant difference in the magnitude of the effects of the 2 muta-tions (Fig 5D, statistical analysis using *LM* and a priori contrasts is given in S6F Fig). Hence, Pol III inhibition has broad health benefits, improving performance of multiple organ systems in old flies, that can be recapitulated by partial loss of function in TFIIIC. This, together with our transcriptomic and lifespan analyses, further implicates tRNAs in late-age health more broadly.

## The effect of partial Pol III inhibition on tRNA expression and proteostatic resilience is conserved in mice

We wanted to examine whether the differential effect of partial loss-of-function in Pol III on tRNA expression was conserved in mammals. We extended our RNA-Seq analysis to mice and used a heterozygous loss-of-function *Polr3B* mutant [15], profiling expression in the superior duodenum, representing a similar tissue to the fly gut, in both female and male mice. Again, we found evidence of differential expression of tRNAs in the $Polr3B^{-/+}$ mutant (Fig 6A and 6B; full analysis results are available in S1 Data). The effect was sexually dimorphic with significant changes observable in females (Fig 6A) but not in males (Fig 6B), echoing the sexual dimor-phism of the lifespan effects observed in flies [12] as well as the sexually dimorphic age-related phenotypes observed in $Polr3B^{-/+}$ mice [15]. Such sexual dimorphism may reflect an underly-ing sex-specific physiology, as is the case for the fly gut [50,51]. Note that fewer differentially expressed tRNAs were detected in mutant mice relative to worms and flies, again consistent with mild phenotypes observed in adult $Polr3B^{-/+}$ mice [15].

In worms and flies, Pol III loss of function and disruptions to tRNA metabolism resulted in increased organismal tolerance of proteostatic stress. To examine whether similar proteostatic resilience can be observed in mice, we isolated dermal fibroblasts and then exposed them to different doses of thapsigargin, another agent that causes ER stress [52]. The dose required to kill 50% of the cells ($LD_{50}$) was significantly higher for both female and male $Polr3B^{-/+}$ mutant fibroblasts compared to wild-type cells (Fig 6C), indicating increased cellular capacity to with-stand proteotoxic stress. Hence, in worms, flies, and mice, a partial loss-of-function in Pol III leads to a disturbance in the composition of the tRNA pool and promotes resistance to proteo-static stress.

## Partial loss of Pol III function impacts translation in worms and flies

Finally, we compared the effect of partial Pol III inhibition on the tRNA pools in different ani-mals. Despite the overall effects on Pol III-target expression appearing similar between species, with dysregulation of tRNAs observed in worms, flies, and mice, we could not find any obvious shared pattern with respect to which tRNAs were differentially expressed. For example, tRNAs belonging to specific isotypes were not consistently affected between species (S7 Fig); a similar

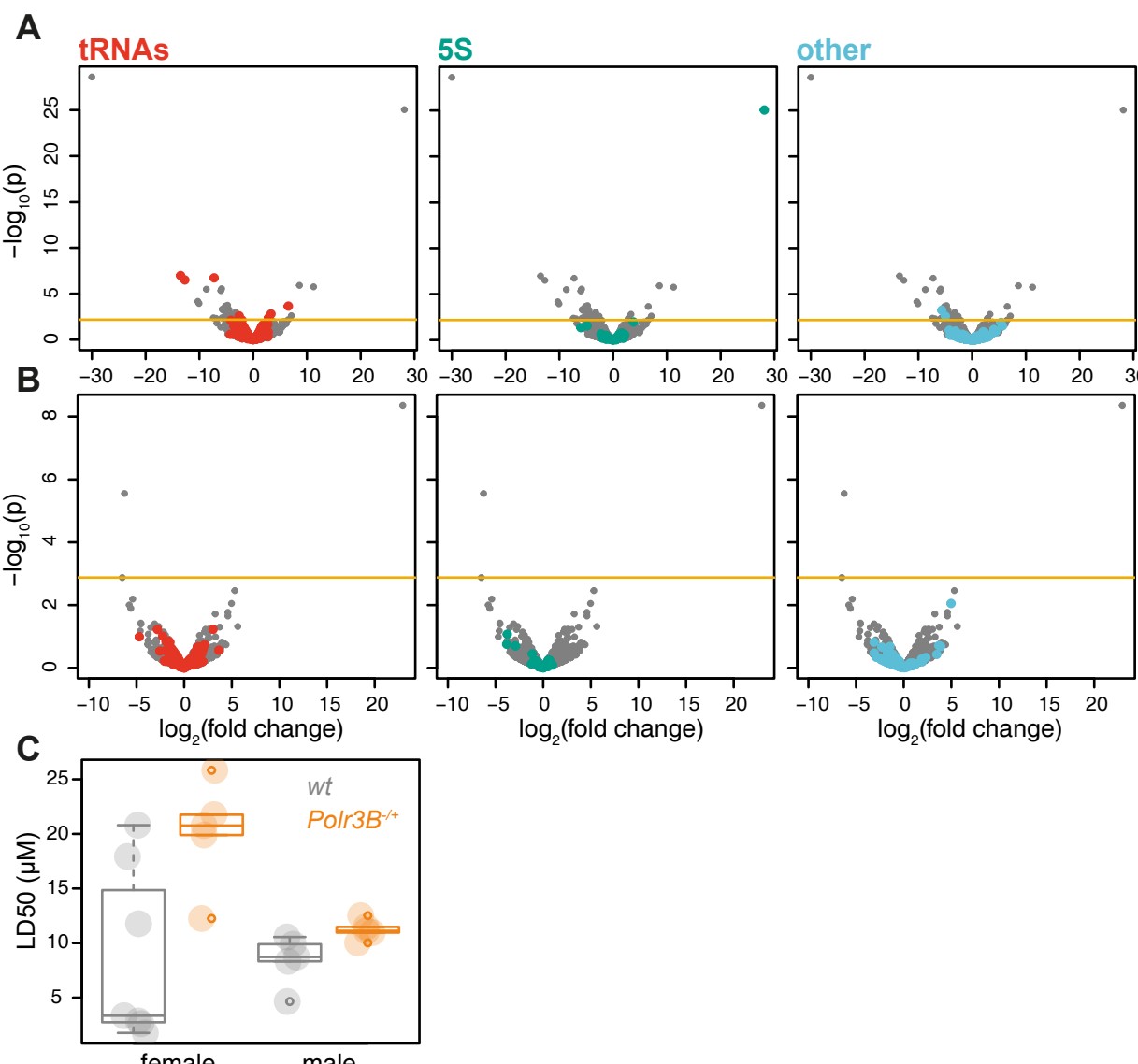

**Fig 6. Partial inhibition of Pol III alters tRNA expression in mice.** Volcano plots of differential expression of short RNAs (<300 b) in the superior duodenum between *Polr3B*<sup>-/+</sup> and wild-type mice, with different classes of transcripts coloured in different panels. (**A**) Female. (**B**) Male. "Other" refers to *7SK*, *RNase P*, *MRP*, *U6*, *Vault*, and *Y* RNA; horizontal line indicates the $p$ value threshold for 10% FDR. (**C**) Thapsigargin LD50 of primary fibroblasts in culture, isolated from female or male, wild-type or *Polr3B*<sup>-/+</sup> mice ($n = 5$–$7$, effect of genotype $p = 6 \times 10^{-3}$, effect of sex or interaction $p < 0.05$, *LM*). Boxplots show quantiles with individual data points overlaid. Data underlying this figure can be found in S1 Data.

case was observed for the isodecoders (S8 Fig). This lack of similarity at the codon level may not be surprising considering the known differences in codon usage and tRNA gene abundance between *C. elegans*, *D. melanogaster*, and mammals [53]. Interestingly, we found that changes in tRNA levels in worms were correlated with the tRNA copy number, while those in flies were correlated with codon usage (S9 Fig). This observation highlights that different forces may shape the final tRNA levels in response to a reduction in Pol III activity, e.g., a substantial feedback is likely occurring in flies to compensate for loss of Pol III function.

Although different between species, the observed tRNA changes could result in altered decoding rates of individual codons, due to changes in abundance of both the cognate tRNA

as well as the competing, near-cognate tRNAs [54]. Using computational models of the ribosomal codon decoding process [55], we confirmed that loss-of-function in Pol III is likely to lead to changes in decoding rates in each organism (Fig 7A–7C). Such alterations to decoding rates could result in changes to both bulk translation as well as to translation of specific mRNAs.

In worms, the translation rates of several codons were predicted to be substantially reduced (Fig 7A). Consistent with this, we observed a significant reduction in overall translation in worms treated with *rpc-1* RNAi using a puromycin-incorporation assay (Figs 7D and S10A). In flies, the predicted change to translation rates were more subtle and balanced (Fig 7B) and indeed we found no evidence for an overall impact of *Polr3D*$^{EY/+}$ on translation in female fly guts (Figs 7E and S10B). To assess whether translation may be qualitatively rather than quantitatively affected in flies, we performed Ribo- and RNA-Seq, again comparing *Polr3D*$^{EY/+}$ to wild-type flies. To obtain sufficient biological material and as a proof-of-principle, we used whole flies (not dissected midguts as above). We sequenced ribosome protected fragments corresponding to over 19,000 open reading frames, even though we recovered a large number of fragments corresponding to rRNA, as is often the case in *Drosophila* [56] (S11 Fig). Combining the Ribo-Seq and RNA-Seq data, we found over 400 mRNAs with evidence of different translation efficiency in the mutant relative to wild-type flies (10% FDR, Fig 7F). Interestingly, this included evidence of increased translation of *Caliban* (*Clbn*) mRNA (Fig 7F), encoding a key component of the Ribosome-associated Quality Control (RQC) pathway that ensures co-translational proteostasis [57,58]. Overall, the tRNA dysregulation observed as a consequence of a partial loss of function in Pol III is predicted to cause a change in translation in worms, flies, and mice, with this prediction experimentally validated in worms and flies. This alteration in translation may be the mechanism underlying the increased proteostatic resilience and longevity.

## Discussion

Our previous work has revealed an evolutionarily conserved role for Pol III in ageing in yeast, worms, and flies [12]. Subsequent assessment of Pol III function indicates that it may also have a role in aspects of ageing and late-life health in mammals [14,15]. Here, we investigated the mechanisms downstream from Pol III that impact longevity by elucidating which of the many Pol III-transcribed genes are responsible for its effects on ageing. By utilising a comparative approach across 3 model organisms, worms, flies, and mice, as well as worm and fly genetics, we identify the tRNAs as the key mediators of the effects of Pol III on longevity. Interestingly, recent work has revealed an enrichment of hypermethylation at tRNA loci in older humans [59], suggesting a role for tRNAs in human ageing.

Across all 3 organisms, tRNA were consistently dysregulated when Pol III activity was reduced, indicating that genes expressed from a type 2 promoter may be more sensitive to the levels of Pol III activity. Additionally, not all tRNAs were equally affected. This may be due to the variation even within the tRNA genes in their ability to recruit Pol III, how easily the loss of their transcription can be compensated and the differences in the cellular requirement for specific tRNAs. This differential sensitivity may not be accidental but may rather point towards latent mechanisms that ensure an appropriate response to a reduction in Pol III activity. At a transcriptional level, such a mechanism is likely to involve chromatin, as Pol III recruitment is modulated in part by local chromatin context [60,61], and it may orchestrate the response of Pol III-transcribed genes, including the tRNA pool, to an absence of growth factors or nutrients. Indeed, physiologically relevant, differential regulation of tRNAs is observed upon manipulation of TORC1 activity in the context of cellular escape from senescence [62].

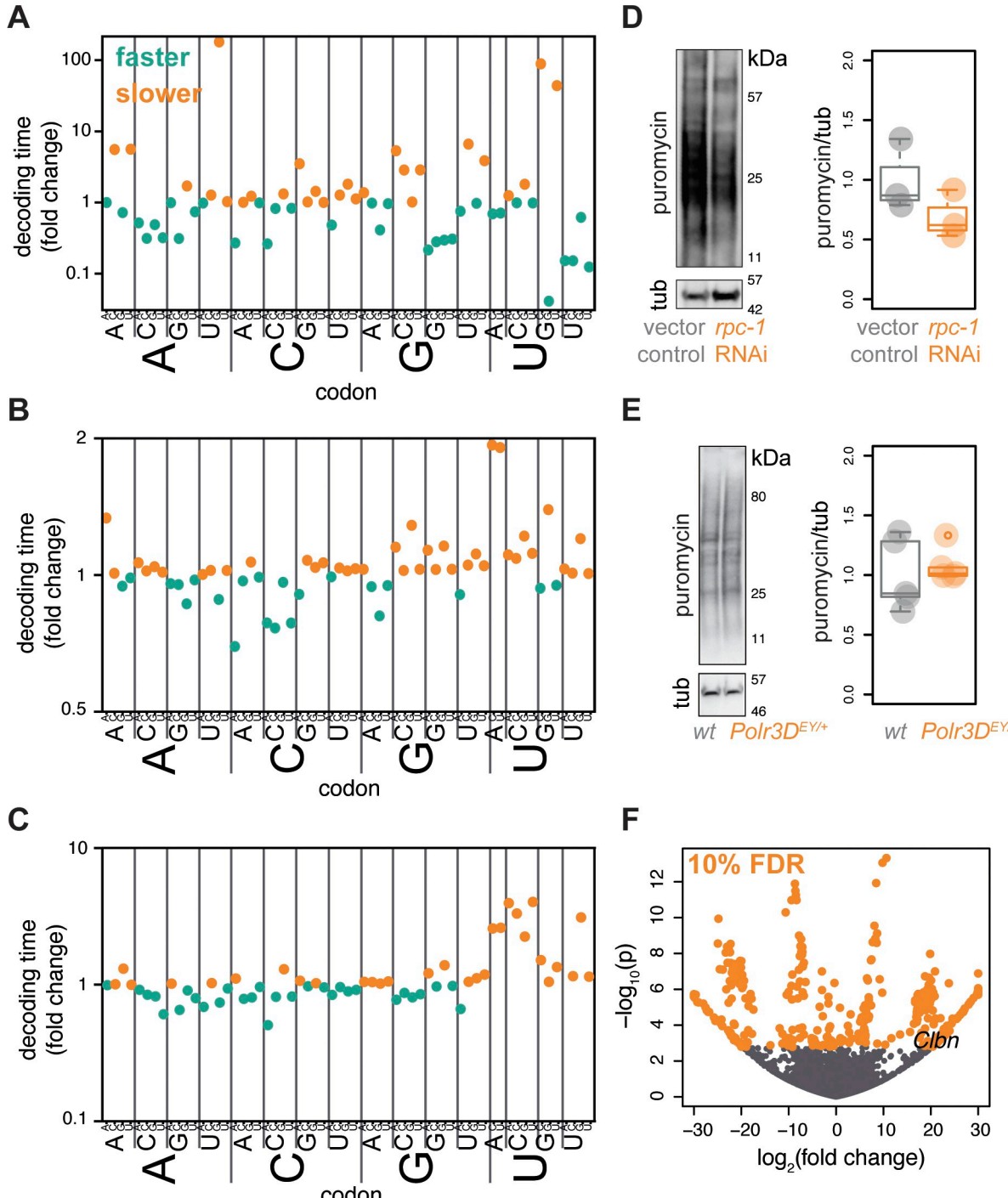

**Fig 7. Predicted and observed changes in translation upon partial inhibition of Pol III across worms, flies, and mice.** Computationally predicted fold changes in codon decoding times resulting from changes in the tRNA pool in animals with a reduced function of Pol III compared to controls, based on the RNA-Seq data. (**A**) *C. elegans*. (**B**) Fruit fly. (**C**) Female mouse. Note differences in Y axes between panels. (**D**) Translation rates measured by puromycin incorporation in worms, example blot and quantification ($n = 3$, $p = 0.033$ *paired t test*). (**E**) Translation rates measured in female fly guts, example blot and quantification ($n = 5$, $p = 0.64$ *paired t test*). Boxplots show quantiles with individual data points overlayed. (**F**) Volcano plot of differential translation efficiency in whole bodies of female *Polr3D*EY/+ flies relative to wild type, estimated from Ribo-/RNA-Seq analysis. Position of *Clbn* is indicated. Data underlying this figure can be found in S1 Data. Uncropped images of blots are presented in S10 Fig.

Furthermore, there is increasing evidence of the role of such differential tRNA expression programmes, and specific tRNAs, in fine-tuning gene expression programmes with consequences for disease [63–66]. Our works extends these findings, revealing the importance of the composition of tRNA pool in organismal ageing.

Interestingly, the pattern of which tRNAs are affected was different in different organisms, hinting that a general disturbance in tRNA pools, and by consequence a disturbance of translation rather than simply a decrease in bulk translation, is promoting longevity. In yeast, deletion of certain tRNA loci results in increased proteostatic resilience, possibly due to an induction of a stress response akin to the induction of UPR observed here by us in a whole-animal context as well as recently by studying mammalian cells in culture [62,67]. Hence, it may not be surprising that mild disruptions to translation, which may occur in various ways, act as a hormetic stress to trigger an activation of other components in the proteostasis-assurance network. The resulting increased proteostatic resilience is likely to contribute to longevity.

Our work brings to the forefront tRNAs as important but hereto unstudied determinants of animal ageing. Their effects on longevity are likely mediated by their effects on translation. Translation is a key component of the network of processes that ensure proteostasis, e.g., translation rates impact co-translational protein folding [68–70], and this is disrupted by ageing [58]. This hypothesis is consistent with previous work by others: translation is controlled at multiple levels and interfering with a number of them, for example, by reducing the provision of translation machinery or rates of translation initiation, promotes longevity (e.g., refs. [47,71–73]). The mechanisms whereby longevity is achieved may not be the same in each case, but in a number of examples including our study there is a triggering of a proteostatic response, suggesting that links between protein translation and proteome health are crucial for healthy ageing [33,74–76]. Our work highlights the role of tRNAs in this interplay. At the same time, it also hints that the mechanisms ensuring proteome health, as indicated by tolerance to proteostatic stress and induction of defensive genes such as BiP orthologues, and those ensuring longevity do not perfectly overlap: in worms we can abrogate different branches of the UPR but maintain the longevity of a Pol III loss-of-function as well as induce UPR without achieving longevity. The explanation for this may rest in the multitude of pathways that ensure proteostasis [77]. It remains possible that one of more pathways not directly examined in this study, such as the RQC pathway, are promoting longevity downstream of Pol III inhibition. At the same time, we should not exclude the possibility that non-canonical roles of tRNAs, i.e., independent of translation, may be involved, as such functions are increasingly described [9,78].

## Materials and methods

### *C. elegans* maintenance and strains

*C. elegans* were routinely grown and maintained on Nematode Growth Media (NGM) seeded with *Escherichia coli* OP50-1. The wild-type strain was Bristol N2. The following strains were used: SJ4005 (*zcIs4 [hsp-4p::GFP]*), SJ17 (*xbp-1(zc12);zcIs4 [hsp-4p::GFP]*), RE666 (*ire-1(v33)*), RB790 (*atf-4(ok576)*), RB772 (*atf-6(ok551)*). JMT200 (*atf-4(ok576);hsp-4p::GFP*) and JMT201 (*atf-6(ok551);hsp-4p::GFP*) were constructed by genetically crossing *zcIs4 [hsp-4p::GFP]* to *atf-4(ok576)* and *atf-6(ok551)*, respectively.

### *C. elegans* lifespan assays

Lifespan assays were performed as previously described [12]. Briefly, synchronous L1 animals were placed on NGM plates seeded with *E. coli* OP50-1 until they reached L4 at 20˚C. At L4 stage, worms were shifted to plates seeded with *E. coli* HT1115 carrying appropriate RNAi

clones obtained from the Ahringer RNAi library. The plates were supplemented with 50 μm FuDR (5-Fluoro-2′-deoxyuridine), unless otherwise specified, and lifespans were carried out at 25°C. For combinatorial RNAi knockdown, the 2 respective RNAi cultures were mixed 1:1 before seeding the plates. In these experiments, controls were also diluted 1:1 with HT115 expressing empty pL4440. Live, dead, and censored worms were recorded every 2 to 3 days by scoring movement with gentle prodding when necessary.

### *C. elegans hsp-4::GFP* expression assays

Synchronous worms expressing the *hsp-4::GFP* transcriptional reporter were raised, treated with RNAi, and aged as for lifespan assay. At specific time points, animals were immobilised on 2% agarose pads using 0.06% Levamisole and imaged immediately using a Leica DMR with 10× zoom. ImageJ software [79] was used for GFP fluorescence quantification. Fluorescence intensity was calculated as pixels per unit area, and background fluorescence subtracted. The values were normalised to those of the control on day 2 of adulthood.

### Worm stress assays

Worms were raised and treated with RNAi as for lifespan assays and assays carried out at 25°C on day 2 adults unless otherwise stated. For heat shock, worms were shifted to 30°C on plates for 3 h. Live, dead, and censored worms were scored daily. For tunicamycin resistance L4 animals were transferred to RNAi plates containing FuDR and supplemented with 40 μg/ml tunicamycin (dissolved in di-methyl sulfoxide, DMSO) or DMSO only. Live, dead, and censored worms were scored daily. For oxidative stress, 100 worms per group were transferred to fresh RNAi plates containing 400 μm of juglone (5-hydroxy-1,4-naphthoquinone). Dead worms were scored every hour until the population died.

### *C. elegans* RNA extraction

Worms were raised and treated with RNAi as for lifespan assays. On day 2 of adulthood worms were washed off the plates with M9 buffer (containing protease inhibitor), allowed to settle and washed again to remove excess bacteria. The tubes containing worms were flash frozen in liquid nitrogen and thawed on ice several times. The worm lysate was centrifuged at 12,000 g for 10 min, supernatant collected in lo-bind eppendorfs and stored in −80°C. To extract total RNA, the worm samples were thawed on ice and supplemented with RNase inhibitor before commencing RNA extraction. Total RNA was extracted using Direct-Zol RNA MiniPrep Kit (Zymo Research) following manufacturer's instructions.

### Fruit fly stocks and husbandry

The wild-type stock was acquired in Dahomey (now Benin) in 1970, and it has been maintained in population cages at 25°C with a 12 h:12 h light/dark cycle. The background used here was cleared of *Wolbachia* several years ago. The $w^{1118}$ or $v^1$ mutations were introduced into this outbred background by extensive backcrossing. The mutations: $CG9609^{G2025}$ (called here $GTF3A^{G2025}$), $l(2)37Cd^{KG04199}$ ($GTF3C5^{KG04199}$), $Pbp49^{KG02425}$ and $Polr3D^{EY22749}$; and transgenes: TIGS, $UAS$-$GTF3A^{RNAi}$ (TRiP.HMS01000), $UAS$-$CG8950^{RNAi}$ (called here UAS-$GTF3C3^{RNAi}$, TRiP.HMC02421) and $UAS$-$Pbp95^{RNAi}$ (TRiP.HMC06384) were backcrossed into the Dahomey background carrying $w^{1118}$ or $v^1$ to allow tracking of the insertion. RNAi lines were from the TRiP project [80]. Flies were reared and experiments performed at 25°C, 12 h:12 h light/dark cycle, 60% humidity, and standard sugar/yeast/agar (SYA) medium [81], unless otherwise noted.

### Fruit fly lifespan assays

Lifespan experiments were carried out as described [12]: progeny of suitable crosses collected over <24 h were reared at standardised larval densities, adults permitted to mate for 48 h after emergence before being sorted by sex on $CO_2$ and placed into plastic vials (15 per vial) in DrosoFlippers (https://www.drosoflipper.com/). Where required, food was supplemented with RU486 (Sigma, M8046) at 200 μm, in which case the control food contained the same volume of the ethanol vehicle. Food was changed and dead/censored flies were recorded trice weekly.

### Fly stress assays

Flies were prepared as for lifespans (female flies, 10 flies per vial). On day 7, the flies were transferred to experimental vials. For tunicamycin resistance flies were introduced onto food containing 5% sucrose and 1.5% agar (no yeast) and tunicamycin (2 mg/l in dimethylsulfoxide; Cell Signalling) on day 9. The equivalent volume of the vehicle alone was added to control treatments (which essentially showed no death in the assay period). Dead flies were scored once or twice daily. For paraquat resistance assays fly food (1% agar, 5% sucrose, and no yeast) was supplemented with 10 mM Paraquat (Sigma, 856177-1G METHYL VIOLOGEN DICHLORIDE HYDRATE, 98%) and dead flies were scored twice daily (beginning and end of the day, approximately 8 h apart).

### Fruit fly gut permeability (smurf) assays

Smurf assays were carried out largely as described [44,47]. Prior to scoring, flies were placed on food containing 2.5% (wt/vol) of blue dye (FD & C blue dye no. 1, Fastcolors, CAS: 3844-45-9) for 48 h. Flies were scored under CO2 as partial smurfs if the dye had seeped outside the gut but had not stained the head or as full smurfs if the entire fly was blue. The same cohort was continuously assayed.

### Fruit fly negative geotaxis (climbing) assays

These were performed essentially as described [47,49]. Flies were transferred to empty vials so that they could climb 2 vial heights in DrosoFlippers (https://www.drosoflipper.com/). Flies acclimatised for 20 to 30 min, were gently tipped to the bottom and the climbing was video recorded. Stills from the same time point (when young wild-type flies start reaching maximum height, most often 15 s) were analysed in Fiji [79] and coordinates exported to Excel (Microsoft). To protect against potential outliers, top 5% and bottom 5% of measurements were removed from all conditions/time points. The same cohort was continuously assayed.

### Fruit fly exploratory walking assay

Exploratory walking assay was performed as described [47,48]. Individual flies were placed in 4 cm diameter/1 cm height circular Perspex arena and video recorded for 15 min, after a 1-min rest. Videos were analysed using Ethovision XT video tracking software (Noldus) to extract the reported parameters. Naive flies were used at each time point.

### Fruit fly RNA extraction and qPCR

Total RNA was quantitatively extracted with Trizol (Invitrogen) from 7- to 9-day-old whole flies (5–10) or dissected midguts (12). cDNA was generated with random hexamers and Superscript II (Invitrogen) according to manufacturer's instructions. Quantitative PCR was performed using Power SYBR Green PCR Master Mix (ABI), on QuantStudio6 Flex real-time

PCR, and quantities of each sequence were determined with the relative standard curve method. Primers used were:

Tubulin F TGGGCCCGTCTGGACCACCA,
Tubulin R TCGCCGTCACCGGAGTCCAT,
Hsc70-4 (BiP) F GCTTGATTGGTCGCAAGT,
Hsc70-4 (BiP) R CTTCTCGTCCTTGTAGGTCA,
pre-tRNAHisGTG-1 F CGTGATCGTCTAGTGGTTAG,
pre-tRNAHisGTG-1 R CCCAACTCCGTGACAATG,
pre-tRNAIleTAT-1 F CGCACGGTACTTATAATCAG,
pre-tRNAIleTAT-1 R, CCAGGTGAGGCTCGAACTC,
pre-tRNALeuCAC-1 F GCGCCAGACTCAAGATTG,
pre-tRNALeuCAC-1 R TGTCAGAAGTGGGATTCG,
U3 forward CACACTAGCTGAAAGCCAAG, and
U3 reverse CGAAGCCCTGCGTCCCGAAC.

The primers specific for pre-tRNAs were designed based on previous biochemical characterization [82] or predicted intronic sequences [83] where each primer set may recognise transcripts emerging from identical or highly similar tRNA genes in different genomic loci.

## Fruit fly ribosome footprinting assay (Ribo-Seq) and analysis

Sample preparation and sequencing for Ribo-Seq and the associated RNA-Seq was performed by CD Genomics, using 300 to 500 mg of whole flies, as described [84] and using RNase I. Library was prepared with NEBNext Multiple Small RNA Library Prep Set for Illumina (catalogue no. E7300S, E7300L) and sequenced using Illumina HiSeq TM X10. Raw sequencing data have been submitted to Gene Expression Omnibus project GSE232724.

FastQC [85] was used to verify the quality of the raw sequencing reads for Ribo-seq and RNA-seq data set, and Cutadapt (v2.10) [86] was used to trim the reads with the following parameters *-a CTGTAGGCACCATCA—error-rate = 0.1—overlap = 3—times = 2*. Ribo-Seq data was assessed with RoboToolkit [87]. Salmon (v0.13.2) [17] was used to quantify the expression of fragments using a whole genome decoy-aware transcriptome index built BDGP6.32 including noncoding RNAs, with parameters—*validateMappings—k 20* for Ribo-Seq—*validateMappings* for RNA-Seq. Subsequent analysis used DESeq2 [18] in R (R version 4.1.2 and 4.2.2) [88]. Initial principal component analysis (PCA) of RNA-Seq data showed that wild-type repeat 3 clustered substantially away from other samples and was therefore removed from further analyses. Differential translation efficacy was assessed as the interaction term in the DESeq2 model: *~ batch (replicate) + condition (genotype) + sequence type (Ribo/RNA) + condition*: *sequence type*. DESeq2 analysis results are given in S1 Data.

## Puromycin incorporation assay

Puromycin incorporation assays were performed essentially as described [12]. Flies were prepared as for lifespans (10 flies per vial). Midguts of 7-day-old female flies were dissected in ice-cold PBS, with wild-type and mutant flies processed in batches carried out in parallel with 10 guts per replicate. Dissected midguts were then transferred to 0.6 ml of ice-cold Schneider's medium (Sigma, #S0146), and 0.4 ml of the same medium pre-warmed to 25°C and supplemented with puromycin (GIBCO, #A11138-02, to a final concentration of 10 mM) was added and incubated at 25°C for 30 min. TCA was added immediately to a final concentration of 12.5%. The midguts were lysed with glass beads (Sigma, #G8772) in a homogeniser (Precellys 24 Ribolyzer). For worms, the protocol was adapted from [89] with minor modifications. Worms were raised on *E. coli* OP50-1 until L4 larval stage. At L4 larval stage, worms were

transferred to control and *rpc-1* RNAi plates. On day 2 of adulthood, exactly 250 worms were picked for each condition and transferred into 100 μl S-basal buffer in a microcentrifuge tube. After 2 washes, worms were incubated in 90 ml S-basal medium supplemented10 ml heat-inactivated *E. coli* OP50-1 and with or without 0.5 mg/ml puromycin. Worms were incubated in microcentrifuge tubes for 3 h at 20°C with gentle manual agitation every 20 to 30 min. Finally, worms were gently centrifuged, washed with S-basal a few times, and homogenised with a pellet pestle in 20 ml SDS loading dye containing proteinase K.

The proteins were separated on gradient gels (Thermo Fisher Scientific, #NP0322BOX). The western blots were probed with anti-puromycin antibody (Milipore, #12D10) followed by HRP conjugated Anti-Mouse secondary (Abcam, #ab6789). Anti-Tubulin antibodies (Sigma, T6199 for flies and T5168 for worms) were used as the loading controls. Blots were imaged in an Amersham ImageQuant800 imager. The intensity of anti-puromycin bands between 15 and 165 kDa was quantified from chemiluminescence images in ImageJ 1.54D [79], relative to tubulin and expressed relative to the control. Batching information was used to perform paired *t* test.

## Northern blot analysis

Relevant RNA sequences were downloaded from FlyBase (https:flybase.org) or WormBase (https:wormbase.org). The sequences were reverse complemented and the best stretch of primer worthy approximately 60 bp in the sequences were identified using Primer3 (https://primer3.ut.ee/). DNA Oligos 3'Digoxigenin (DIG)-labelled and HPLC purified were ordered from Eurofins Genomics (see below). Midguts of the 7-day-old female flies were dissected and total RNA was quantitatively isolated as described above. For worms, total mRNA was isolated from day 2 adults as described for small RNA-seq, without the size selection step. The RNA were separated in 6% TBE-Urea gel (Invitrogen, EC68652BOX) and transferred to BrightStar-Plus Positively Charged Nylon Membrane (AM10100). The RNA was fixed with UV-crosslinking in (Stratagene UV Stratalinker 1800). The probes (25 pmole) were hybridised at 64°C overnight in Roche DIG Easy Hyb Hybridization Buffer (11603558001) with gentle agitation. The membrane washing and blocking were performed with Roche DIG Wash and Block Buffer Set (11585762001, version 12). Hybridised DIG-labelled probes were detected with DIG Roche Anti-Digoxigenin-AP, Fab fragments (11093274910) and Roche CDP-Star, ready-to-use (12041677001) reagent as per manufacturer's instructions. The membranes were imaged and quantified as described under puromycin incorporation assay. Actin was used as the loading control. Probes used for Northern Blotting are as follows:

Flies

tRNA Glutamine-TTG-1-1 FBtr0076083 GATTTGAACTCGGATCGCTGGATTCAAAG TCCAGAGTGCTAACCATTACACCATAGAACC

tRNA Histidine-GTG-1-1 FBtr0087963 GCCGTGACCAGGATTCGAACCTGGGTTACCACGGCCACAACGTGGGGTCCTAA CCACTAG

Actin (Act57B) FBgn0000044 ACGGGGAGCGTCATCACCGGCGAAACCGGCCTTGCACATGCCGGAGCCATTGT CAACGAC

Worms

tRNA Leucine- TAA2.1 T22C8.t1 GTTCGAACCCACGCGGTGTTATCCACCATTGGAACTTAAGGCCAACGCCTTAAC CACTCG

tRNA Arginine- TCG-2-1 F14B4.t1

GATCGAACCTGTAATCCCCAGATTCGAAGTCTGGTGGCTTATCCATTAGGCGAC
GCGACC

Actin (act-1) T04C12.6
GGTGACACCATCTCCAGAGTCGAGGACGACTCCGGTGGTACGTCCGGAAGCGT
AGAGGGA

## Mouse lines and husbandry

The mouse line C57BL/6N-Polr3b<em7(IMPC)Tcp>/Tcp was provided as part of the Nor-
COMM2 project funded by Genome Canada and the Ontario Genomics Institute (OGI-051) at
the Toronto Centre for Phenogenomics. This allele from project TCPR0237 was generated by
injecting Cas9 D10A mRNA and a single guide RNA with the spacer sequence TCATGTCCCT-
CAGACGGCAC. This resulted in an 8 bp insertion TCCAGTGT at Chromosome 10 positive
strand 84632456 bp (GRCm38) predicted to cause a frameshift mutation with early truncation;
c.305_306insTCCAGTGT; p.R103Pfs*3. The generation of founder mice was performed in com-
pliance with the Animals for Research Act of Ontario and the Guidelines of the Canadian Council
on Animal Care. The TCP Animal Care Committee reviewed and approved all procedures con-
ducted on animals at TCP. Four male and 5 female 2-month-old heterozygous C57BL/
6N-Polr3b<em7(IMPC)Tcp>/Tcp (hereafter Polr3b$^{+/-}$) founder mice were obtained and bred
in-house with non-transgenic C57Bl/6N mice. To avoid any possible carry-over effect in wild-
type offspring derived from Polr3b$^{+/-}$ parents, experimental mice were generated by mating
Polr3b$^{+/-}$ or Polr3b$^{+/+}$ offspring with wild-type C57Bl/6N mice to generate experimental Polr3b$^{+/-}$
or Polr3b$^{+/+}$ mice, respectively [15]. Mice were maintained as previously described [90], under a
UK Home Office Project Licence (PDBDC7568), following local ethical review, and adherance to
the "principles of laboratory animal care" (NIH Publication No. 86–23, revised 1985).

## Mouse genotyping

Mice were genotyped from ear biopsies either in-house by PCR, or by qPCR using an external
provider (Transnetyx, Cordova, Tennessee, United States of America). In-house PCR was car-
ried out on genomic DNA using primer pairs specific for the wild-type allele (for: 5′-AGGCT
GCTGTGCACTGTATT-3′; Rev: 5′-GACGGCACTGGAGCAGAAT-3′; band size = 82 bp) or
the mutant allele (for: 5′-TCAGTGGGGAAAGTTCAGGC-3′; Rev: 5′-TCAGACGGACACT
GGACACT-3′; band size = 110 bp).

## Mouse dermal fibroblasts cultures and thapsigargin survival assays

Primary dermal fibroblasts from 4- to 6-week-old wild-type and *Polr3b$^{+/-}$* mice were isolated
and cultured as previously described [91], and then cellular resistance to endoplasmic reticulum
stress was assessed as previously described [52]. In brief, fibroblasts were exposed to thapsigar-
gin (0–100 μm; Sigma Aldrich (Dorset, United Kingdom)) and subsequently incubated at 37˚C
for 24 h in DMEM. Following incubation, cells were washed with 1× PBS, incubated in DMEM
supplemented with Pen-Strep, L-Glutamine, and 10% FBS at 37˚C. Cell viability was measured
using the WST-1 test following the manufacturer's protocol (Roche Diagnostics, West Sussex,
UK). Each cell line was exposed to thapsigargin in triplicate and LD$_{50}$, the dose which caused
50% cell death, calculated using probit analysis (AAT Bioquest, California, USA).

## Mouse RNA extraction

Following an overnight fast, mice were culled by cervical dislocation and then duodenum was
dissected, flushed with PBS, immediately frozen in liquid nitrogen, and stored at −80˚C. RNA

was subsequently isolated following homogenisation with Trizol (Invitrogen, Paisley, UK) following manufacturer's instructions.

### RNA-Seq of Pol III-transcribed genes

The RNA samples obtained from worm, fly, and mouse were selected for RNAs less than 200 nucleotides in length using mirVANA RNA isolation kit. The final elute containing the small RNAs was demethylated using rtStar tRF&tiRNA Pretreatment Kit following manufacturer's instructions. Finally, the RNA cleanup was performed using Zymo-RNA Clean and Concentrator-5 cleanup. RNA quality was checked prior to Library preparation using Agilent Small RNA kit for Bioanalyzer. The typical yield at this step was 4 to 6 μg of <200 nucleotide-long, demethylated, DNA-free RNA in 8 μl of water. Library preparation and sequencing was done by Glasgow Polyomics, Glasgow, UK. Briefly, libraries were prepared using TruSeq Nano kit, analysed using Bioanlyser and sequenced on the Nextseq using paired end 2*75 bp read length to a sample average of 15 M reads. Raw sequencing data have been submitted to Gene Expression Omnibus projects: GSE232719, GSE232720, GSE232721, and GSE232723.

For each data sets, raw sequencing reads were checked using FastQC [85] and trimmed by Cutadapt (v2.10) [86] using default parameters. Transcript abundance were quantified using Salmon (v0.13.2) [17] with parameters—*validateMappings*, using all-RNA, decoy-aware transcriptome indexes built from WBcel235, GRCm39, or BDGP6.32 including noncoding RNAs for worm, mouse, and fly, respectively. The data for sequences shorter than 300 bases were imported into R with Tximport [92], and differential expression analysis performed with DESeq2 using a standard workflow [18]. Genomic tRNA Data Base [83] was used to obtain information on tRNAs for each organism. tRNA gene occurrence in the genome was counted based on files in the tRNA Data Base [93]. Codon usage per species was obtained from Codon Usage Data Base [94]. DESeq2 analysis results are given in S1 Data.

Codon decoding times were predicted using Gillespie models of the tRNA-dependent codon decoding process [95,96]. Modified bases for tRNA anticodons were retrieved from the tRNAdb2009 database [97] and individual tRNA: codon relationships were classified as Watson–Crick decoding cognate, wobble-decoding cognate, near-cognate, or non-cognate using base-pairing schemes as described [95]. Cellular concentrations for each tRNA were calculated assuming a total cellular tRNA concentration of 190 μm [95] and using the fractional signal for each tRNA in the RNA-Seq data. Total concentrations of the 4 different tRNA classes for each codon were then calculated by summing the concentration for all tRNAs in each class. Codon decoding times were estimated by averaging results from 10,000 repeat runs of the Gillespie model for each codon, using the tRNA class concentrations as input.

### Statistical analysis

Survival differences were assessed by a log-rank test either using the online freeware OASIS (http://sbi.postech.ac.kr/oasis/surv/), in the case of worm survival, or Excel, in case of fly survival. Linear models and ordinal logistic regression were performed either in R or in JMP Pro 17.0.0 and each model had a full factorial design. ANOVA, post hoc tests, and *t* tests were performed in JMP Pro 17.0.0.

### Supporting information

**S1 Table. Demography and statistical analysis of survival with Tunicamycin or after heat-shock.** In each case trial A is presented in Fig 2.
(DOCX)

**S2 Table. Demography and survival analysis for worm juglone resistance and lifespans.** *P* value is from the log-rank test to RNAi control. For each set of experiments, trial A is presented in S1 Fig.
(DOCX)

**S1 Fig. Northern blots on worms treated with *rpc-1* RNAi.** Full images of northern blots presented in Fig 1B. Note that the *Actin* band corresponding to 1 wild-type sample was accidentally lost from the blot.
(TIF)

**S2 Fig. Inhibition of Pol III in worms with a loss-of-function in UPR TFs.** (**A**) Survival of control or *xbp-1* RNAi worms exposed to juglone. (**B**–**D**) Expression of the *hsp-4::GFP* reporter in *rpc-1* RNAi- or vector control-treated worms that were either wild type or carried mutations in *xbp-1*, *atf-4* or *atf-6*. *LM* analysis with a fully factorial design (age, genotype, RNAi) showed significant effects of all covariates and all interactions ($p < 0.05$) for all 3 experiments. The letters on top of the graphs indicate the results of the *Tukey–Kramer HSD test*, where outcomes that are not connected by a letter within an experiment are significantly different ($p < 0.05$). Boxplots show quantiles with individual data points overlayed. (**E, F**) Lifespans of control or *xbp-1* or *atf-4* mutant worms treated with control or *rpc-1* RNAi. (**G**) Lifespans of worms treated with *atf-6* RNAi, *rpc-1* RNAi, or both. (**H**) Lifespans of worms treated with *rpc-1* RNAi or RNAi against individual aminoacyl tRNA synthases whose knockdown was able to induce *hsp-4::GFP*. For survival and lifespans, demography, statistics and additional repeats are shown in S2 Table. Data underlying this figure can be found in S1 Data.
(TIF)

**S3 Fig. Quantifications of tRNA and pre-tRNA levels in flies.** (**A** and **B**, **C** and **D**) Full images of blots presented and quantified in Fig 3B and 3C. (**E**) Relative *pre-tRNA*$^{Ile}$ levels ($n = 7$–$8$, $p = 0.089$ *t* test). (**F**) Relative *pre-tRNA*$^{Leu}$ levels ($n = 7$–$8$, $p = 0.88$ *t* test). Boxplots show quantiles with individual data points overlayed. Data underlying this figure can be found in S1 Data.
(TIF)

**S4 Fig. Lifespans of female flies with P-element insertions in TFIIIA, TFIIIC, and SNAPc, paraquat survival.** (**A**) Wild-type $n = 133$ dead/17 censored flies, *GTF3A*$^{G/+}$ $n = 119/13$, $p = 6.6 \times 10^{-6}$. (**B**) Wild-type $n = 64/0$, *GTF3A*$^{G/+}$ $n = 73/1$, $p = 0.41$. (**C**) Wild type $n = 148/2$, *GTF3A*$^{G/+}$ $n = 145/6$, $p = 0.88$. (**D**) Wild-type $n = 109/42$, *GTF3A*$^{G/+}$ $n = 117/38$, $p = 9.9 \times 10^{-7}$. (**E**) Wild-type $n = 68/0$, *GTF3A*$^{G/+}$ $n = 73/1$, $p = 0.41$. (**F**) Wild-type $n = 124/7$, *Pbp49*$^{KG/+}$ $n = 140/2$, $p = 0.50$. Note this is part of the same experiment as in Fig 4A so the wild-type survival is replotted. (**G**) Wild-type $n = 170/0$, *Pbp49*$^{KG/+}$ $n = 137/0$, $p = 9.5 \times 10^{-16}$; *GTF3C5*$^{KG/+}$ $n = 150/0$, $p = 1 \times 10^{-3}$. (**H**) Wild-type $n = 151/0$, *Pbp49*$^{KG/+}$ $n = 147/2$, $p = 0.11$; *GTF3C5*$^{KG/+}$ $n = 134/8$, $p = 9.6 \times 10^{-4}$. (**I**) Wild-type $n = 79/0$, *Pbp49*$^{KG/+}$ $n = 69/3$, $p = 0.75$; *GTF3C5*$^{KG/+}$ $n = 79/0$, $p = 2.3 \times 10^{-3}$. (**J**) Survival of paraquat exposure (wild-type $n = 100/0$, *GTF3A*$^{G/+}$ $n = 100/0$, $p = 0.059$). All $p$ values are from *log-rank test* to wild type. Data underlying this figure can be found in S1 Data.
(TIF)

**S5 Fig. Pre-tRNA expression in TFIIIC mutants and lifespans of female flies with the inducible TIGS driver.** (**A**) Relative *pre-tRNA*$^{His}$ levels ($n = 6$–$7$, $p = 0.88$ *t* test). (**B**) Relative *pre-tRNA*$^{Ile}$ levels ($n = 7$, $p = 0.15$ *t* test). (**C**) Driver alone control (-RU486 $n = 125$ dead/25 censored flies, +RU486 $n = 133/11$, $p = 0.28$, *log-rank test*). (**D**) *TIGS>GTF3A*$^{RNAi}$ females (-RU486 $n = 149/2$, +RU486 $n = 149/3$, $p = 6.4 \times 10$–$16$, *log-rank test*). (**E**) *TIGS>Pbp95*$^{RNAi}$

females (-RU486 $n$ = 131/15, +RU486 $n$ = 148/2, $p$ = 0.027, *log-rank test*). Data underlying this figure can be found in S1 Data.
(TIF)

**S6 Fig. Health phenotypes.** (**A**) Results of statistical analysis of loss of gut barrier function (smurf assay) on *Polr3D$^{EY/+}$*, *GTF3C5$^{KG/+}$* and wild-type females using *ordinal logistic regression*. (**B**) *P* values from the multivariate *LM* analysis. (**C**) Moving duration, (**D**) velocity, and (**E**) rotation frequency during exploratory walking by *Polr3D$^{EY/+}$* and wild-type females. Boxplots show quantiles with individual data points overlayed. (**F**) Results of statistical analysis of negative geotaxis assay (climbing assay) on *Polr3D$^{EY/+}$*, *GTF3C5$^{KG/+}$* and wild-type females using *LM*. In all analysis, age is given in days. Data underlying this figure can be found in S1 Data.
(TIF)

**S7 Fig. Isotype tRNA level changes across species.** Differential expression (log$_2$ fold change) for tRNAs grouped based on the amino acid they decode in worms, flies (gut), and female and male mice (duodenum) obtained from RNA-Seq analyses. Amino acids are given in single-letter code with Mi indicating initiator tRNA$^{Met}$, with tRNAs differentially expressed upon loss of function of Pol III in each species indicated in red (10% FDR). Note differences in Y axes between panels. Data underlying this figure can be found in S1 Data.
(TIF)

**S8 Fig. Isodecoder tRNA level changes across species.** Differential expression (log2 fold change) for tRNAs grouped based on the anticodon in worms, flies (gut), and female and male mice (duodenum) obtained from RNA-Seq analyses. tRNAs differentially expressed upon loss of function of Pol III in each species are indicated in red (10% FDR). Note differences in Y axes between panels. Data underlying this figure can be found in S1 Data.
(TIF)

**S9 Fig. Relationships between observed tRNA abundance changes and tRNA occurrence in the genome or codon usage.** (**A**) Worms. (**B**) Flies. (**C**) Female mice. * Indicates $p < 0.05$ *LM*. Data underlying this figure can be found in S1 Data.
(TIF)

**S10 Fig. Translation rates.** Full blots of quantification of puromycin incorporation in (**A**) worms and (**B**) female fly guts. Paired samples that were obtained at the same time are placed next to each other on the blot. Quantifications are given in Fig 7.
(TIF)

**S11 Fig. Ribo-Seq.** Mappings to types of RNA and open reading frames (ORFs) for unique or all ribosome protected fragments (RPF) in 3three replicate samples of wild-type or mutant (*Polr3D$^{EY/+}$*) female flies (whole body). Data underlying this figure can be found in S1 Data.
(TIF)

**S1 Raw Images. Raw blot images.** These are also presented in the relevant supplementary figures.
(PDF)

**S1 Data. This Excel file contains data presented in the manuscript.**
(XLSX)

## Acknowledgments

Some *C. elegans* strains were provided by the CGC, which is funded by NIH Office of Research Infrastructure Programs (P40 OD010440). The TIGS fly line was a kind gift of L. Partridge and S. Pletcher. Stocks obtained from the Bloomington Drosophila Stock Center (NIH P40OD018537) were used in this study and the RNAi lines were made by the TRiP project (Office of the Director R24 OD030002: "TRiP resources for modelling human disease," PI: N. Perrimon). We thank the members of our laboratories and institutes for support during this study.

## Author Contributions

**Conceptualization:** Susan Broughton, Tobias von der Haar, Colin Selman, Jennifer M. A. Tullet, Nazif Alic.

**Formal analysis:** Yasir Malik, Yavuz Kulaberoglu, Shajahan Anver, Sara Javidnia, Gillian Borland, Rene Rivera, Stephen Cranwell, Danel Medelbekova, Tatiana Svermova, Jackie Thomson, Susan Broughton, Tobias von der Haar, Colin Selman, Jennifer M. A. Tullet, Nazif Alic.

**Funding acquisition:** Colin Selman, Jennifer M. A. Tullet, Nazif Alic.

**Investigation:** Yasir Malik, Yavuz Kulaberoglu, Shajahan Anver, Sara Javidnia, Gillian Borland, Rene Rivera, Stephen Cranwell, Danel Medelbekova, Tatiana Svermova, Jackie Thomson, Susan Broughton.

**Methodology:** Yasir Malik, Yavuz Kulaberoglu, Shajahan Anver, Sara Javidnia, Gillian Borland, Rene Rivera, Stephen Cranwell, Danel Medelbekova, Tatiana Svermova, Jackie Thomson, Susan Broughton.

**Project administration:** Colin Selman, Jennifer M. A. Tullet, Nazif Alic.

**Supervision:** Yasir Malik, Yavuz Kulaberoglu, Shajahan Anver, Sara Javidnia, Gillian Borland, Colin Selman, Jennifer M. A. Tullet, Nazif Alic.

**Validation:** Yasir Malik, Yavuz Kulaberoglu, Shajahan Anver, Sara Javidnia, Gillian Borland, Rene Rivera, Stephen Cranwell, Danel Medelbekova, Tatiana Svermova, Jackie Thomson, Susan Broughton.

**Visualization:** Yasir Malik, Shajahan Anver, Sara Javidnia, Gillian Borland, Rene Rivera, Stephen Cranwell, Danel Medelbekova, Tatiana Svermova, Jackie Thomson, Susan Broughton, Tobias von der Haar, Colin Selman, Jennifer M. A. Tullet, Nazif Alic.

**Writing – original draft:** Yasir Malik, Yavuz Kulaberoglu, Shajahan Anver, Gillian Borland, Tobias von der Haar, Colin Selman, Jennifer M. A. Tullet, Nazif Alic.

**Writing – review & editing:** Yasir Malik, Yavuz Kulaberoglu, Shajahan Anver, Gillian Borland, Rene Rivera, Stephen Cranwell, Danel Medelbekova, Tatiana Svermova, Jackie Thomson, Susan Broughton, Tobias von der Haar, Colin Selman, Jennifer M. A. Tullet, Nazif Alic.

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
