## [Editor Report · Decision Letter 0]

10 Oct 2023

Dear Dr Alic, 

Thank you for submitting your manuscript entitled "Disruptions to tRNA biogenesis enhance proteostatic resilience, improve later-life health and promote longevity" for consideration as a Research Article by PLOS Biology.

Your manuscript has now been evaluated by the PLOS Biology editorial staff as well as by an academic editor with relevant expertise and I am writing to let you know that we would like to send your submission out for external peer review.

Once your full submission is complete, your paper will undergo a series of checks in preparation for peer review. After your manuscript has passed the checks it will be sent out for review. To provide the metadata for your submission, please Login to Editorial Manager (https://www.editorialmanager.com/pbiology) within two working days, i.e. by Oct 12 2023 11:59PM.

Kind regards,

Ines

--

Ines Alvarez-Garcia, PhD

Senior Editor

PLOS Biology

---

## [Decision Letter · Decision Letter 1]

21 Dec 2023

Dear Dr Alic,

Thank you for your patience while your manuscript entitled "Disruptions to tRNA biogenesis enhance proteostatic resilience, improve later-life health and promote longevity" was peer-reviewed at PLOS Biology. Please also accept my apologies for the delay in providing you with our decision. The manuscript has now been evaluated by the PLOS Biology editors, an Academic Editor with relevant expertise, and by three independent reviewers. 

As you will see, the reviewers find the conclusions interesting and novel, but they have also raised several concerns that should be addressed before we can consider the manuscript for publication. After discussing all the concerns with the Academic Editor, we would like to highlight the following points for you to address, although we should note that other comments from all 3 reviewers will need to be addressed as well, with additional experiments or changes to the text as appropriate:

1. Can you effectively distinguish "the effect of pol III inhibition on proteostatic resistance from its overall effect on lifespan"? This was raised by Reviewers 1 and 2 and it will be important to address/clarify it.

2. Several concerns have been raised regarding the inconsistency in tRNA levels across species under the experimental conditions (e.g. pol III inhibition). An orthogonal method such as northern blotting is needed to confirm a number of the changes in tRNA levels, given that misleading results could plausibly occur from RNA-seq given that tRNAs are highly modified. Although you have completed a demethylation step, this is sufficient and a subset of tRNAs should be chosen for additional confirmation and to verify the differential effects of pol3 inhibition in worms vs. flies. In addition, Reviewer 2 noted "the surprising observation that many tRNAs are upregulated in heterozygous Polr3d mutants" and this result should be verified with an orthogonal method as well. 

3. It will be "important to provide information about global translation rates in pol III-inhibited vs control organisms or cell lines", with a comparison between worms and flies as a priority, as suggested by Reviewer 3. In addition, this reviewer thinks it is also important assessing translation.

In light of the reviews and discussion with the Academic Editor, we would like to invite you to revise the work to thoroughly address the reviewers' reports. Given the extent of revision needed, we cannot make a decision about publication until we have seen the revised manuscript and your response to the reviewers' comments. Your revised manuscript is likely to be sent for further evaluation by all or a subset of the reviewers.

**IMPORTANT - SUBMITTING YOUR REVISION**

3. Resubmission Checklist

a) *PLOS Data Policy*

b) *Published Peer Review*

Sincerely,

Ines

--

Ines Alvarez-Garcia, PhD

Senior Editor

PLOS Biology

Reviewers' comments

Rev. 1:

This study by Malik et al looks to further explore how inhibition of RNA polymerase III promotes longevity. They focus on the impact of RNA Pol III on disrupting tRNA levels across species including in worms, flies, and mice. They further find that disruption of tRNA synthases is sufficient to induce an unfolded protein response reporter. Furthermore they find that disruption of the TFIIIC transcription factor, which is important for tRNA expression, extends lifespan in flies. Overall I believe this to be an advancement for the field of aging biology by providing new insights into the role tRNAs and RNA Pol III in the aging process.

Major point: The paper primarily focuses on the effects of RNA Pol III disruption being potentially driven through changes in tRNA levels including lifespan extension. While the authors show that disruption of tRNA synthases can induce the UPR, does this also extend lifespan? Was this tested and just not shown? It was shown in yeast and worms that tRNA synthetase inhibitors extend lifespan (Robbins et al 2022), it would further strengthen the authors arguments if specifically disrupting tRNA synthases extended lifespan.

Other points:

Some of the data is not as convincing as the conclusions made.

On page 13 - Similarly, Polr3DEY/+ delayed age-related loss in climbing ability (Figure 5C). The p-value given is (n=99-126, effect of genotype p=4.7x10-6, age p<2x10-16, age by genotype interaction p=2.7x10-6, LM) and yet the data does not look so clear. At 4 and 11 days Polr3D EY/+ appears to have lower climbing ability to the WT. Then only at 18 days the mutant may have slightly higher climbing ability, but this is highly overlapping with WT, and then at 25 days the mutant and WT look essentially identical.

Minor points - its unclear why the authors use shades of yellow-orange and orange-red in some of their graphs, as these colors are sometimes harder to distinguish - Fig 5 and especially Sup Fig 7.

Rev. 2: Alessandro Vannini – note that this reviewer has signed his review

In this manuscript, Malik et al show that across species pol III inhibition preferentially alters the levels of a subset of tRNAs, which consistently results in extended lifespan by improving resistance to proteostatic stress. This work is a continuation of the important research done in the authors'laboratories that has uncovered a connection between pol III activity and ageing in several species. While the link between tRNA levels, ageing, and the unfolded protein response (UPR) presented in this manuscript make for an intriguing model, we find the details of these connections to be possibly underexplored.

In particular, we think that the model proposed in this manuscript could be strengthened by addressing the following points:

- It is not clear whether the effect of pol III inhibition on proteostatic resistance can be effectively distinguished from its overall effect on lifespan. For example, in figure 2A, the shift in survival of rpc-1 RNAi vs vector control worms (orange vs grey solid lines) is interpreted as a sign of increased resistance to tunicamycin. However, rpc1 knockdown generally extends the lifespan of worms and in fact a similar shift can be observed for worms not exposed to tunicamycin (orange vs grey dotted lines). It is therefore difficult to interpret this as a specific effect on resistance to proteostatic stress. It would be useful if the authors could show the survival data from tunicamycin-treated samples/organisms normalised by their non-tunicamycin-treated counterparts. The same applies to figure 2B and figure 4D.

- Given that there doesn't seem to be any consistency in how specific tRNAs are affected by pol III inhibition across species, and that other mechanisms that affect translation rates are also known to prolong longevity, the authors propose that the effects of tRNA levels on ageing are likely mediated by their effect on translation rates. While we find this hypothesis convincing (generally lower translation rates could indeed explain both the extension in life span and the increase in tolerance of proteostatic stress), it would be important to provide information about global translation rates in pol III-inhibited vs control organisms or cell lines. In particular, it would be interesting to compare differences in translation rates in worms, where pol III inhibition mostly causes a downregulation of tRNA levels, to those of flies, where surprisingly the levels of most tRNA seem to be increased upon pol III inhibition. Are global translation rates affected similarly in both cases, despite levels of tRNAs changing in opposite directions? If this is not the case, how can almost opposite effects on tRNA levels be linked to the same downstream outcome on lifespan?

- Again related to the surprising observation that many tRNAs are upregulated in heterozygous Polr3d mutants (Fig 3A), it is not clear to us how the results from these organisms, namely the increased expression of BiP (Fig 3B), relate to the data from flies in which recruitment to type II promoters is directly affected (Fig 4). Are tRNA levels also counterintuitively upregulated in GTF3C5 mutants? If they are, as expected, mostly downregulated, this would effectively uncouple the results from figures 3 and 4.

- In supplementary figure 1 A-C, the authors report the effects of rpc-1 knockdown in wild type worms and worms carrying mutations in each of the three UPR branches. The plots show that in all mutants the general levels of expression are the reporter genes are much lower than in the wild type, independently of the life stage or RNAi treatment. This experiment does not then indicate that all three branches of the UPR are engaged following Pol III inhibition, as the authors conclude, but that mutations in each of the branches are sufficient to completely hinder expression of the reporter gene. It would be useful if the authors could perform similar experiments using a different, possibly endogenous, read-out of the UPR, which is not completely impaired by each of the mutations.

- In the discussion, the authors state that the variation seen in the effect of Pol III inhibition on different tRNAs might be due, among other things, to a difference in tRNA gene copy numbers. This is an interesting hypothesis that could be tested with just some additional analysis of the data presented in this manuscript, by checking if in each species tRNA genes with higher copy numbers are less affected by pol III inhibition and/or vice versa.

Rev. 3:

This article by Malik et al. uses several model organisms to better define how pol3 activity is linked to longevity. Interestingly, they observe that tRNAs likely underpin the complex physiological effects, at least in part through activation of the UPR. The unifying theme across model organisms adds credibility to the hypothesis, although the specific effects on tRNAs are distinct in flies and mice. Although female-specific changes are noted, I think the article would benefit from further discussion and speculation about the sex-specific differences. It would be important in future work to determine how translation is affected and whether specific sets of transcripts are sensitive to the tRNA changes. The authors allude to this possibility but addressing this question in a rigorous way might be beyond the scope of this paper.

Minor note: In the Figure 2 title, "metabolisms" should be replaced with "metabolism"

---

## [Decision Letter · Decision Letter 2]

23 Aug 2024

Dear Dr Alic,

Thank you for your patience while we considered your revised manuscript entitled "Disruptions to tRNA biogenesis enhance proteostatic resilience, improve later-life health and promote longevity" for publication as a Research Article at PLOS Biology. This revised version of your manuscript has been evaluated by the PLOS Biology editors, the Academic Editor and two of the original reviewers.

Based on the reviews, we are likely to accept this manuscript for publication, provided you satisfactorily address the data and other policy-related requests stated below.

In addition, we would like you to consider a suggestion to improve the title:

“Disruption of tRNA biogenesis enhances proteostatic resilience, improves late-life health and promotes longevity"

We expect to receive your revised manuscript within two weeks. 

*Published Peer Review History*

*Press*

Sincerely,

Ines

--

Ines Alvarez-Garcia, PhD

Senior Editor

PLOS Biology

DATA POLICY:

Thank you for submitting a file containing all the data underlying the graphs shown in the figures. I have checked the data and it's not clear to me which data corresponds to the following figures in the file - please clearly label the data accorind to the figures to make this easier to readers:

Fig. 1A; Fig. 3A; Fig. 6A, B; Fig. 7A-C, F; Fig. S7 and Fig. S8 

Please also ensure that figure legends in your manuscript include information on WHERE THE UNDERLYING DATA CAN BE FOUND.

** In addition, we ask you to make publicly available at this stage the data you have deposited at the GEO database: GSE232719, GSE232720, GSE232721, GSE232723 and GSE232724

CODE POLICY

Reviewers' comments

Rev. 2:

In the manuscript "Disruptions to tRNA biogenesis enhance proteostatic resilience, improve later-life health and promote longevity", Dr. Nazif Alic and colleagues expand on their previous findings regarding RNA polymerase III activity and aging. In this manuscript the authors convincely demonstrate a direct role of tRNAs level in ageing across different species, this highlighting this as important evidence conserved throughout evolution. The manuscript is very well written and easy to follow. It will be appreciated by a broad readership. The experiments are well thought and well controlled providing good evidence to substantiate the author's claims. This is an important manuscript and I feel it can be published in its current form.

Rev. 3:

I thank the authors for their responses. I have no further comments.

---

## [Editor Report · Decision Letter 3]

20 Sep 2024

Dear Dr Alic,

Thank you for the submission of your revised Research Article entitled "Disruption of tRNA biogenesis enhances proteostatic resilience, improves later-life health and promotes longevity" for publication in PLOS Biology. On behalf of my colleagues and the Academic Editor, Dylan Taatjes, I am delighted to let you know that we can in principle accept your manuscript for publication, provided you address any remaining formatting and reporting issues. These will be detailed in an email you should receive within 2-3 business days from our colleagues in the journal operations team; no action is required from you until then. Please note that we will not be able to formally accept your manuscript and schedule it for publication until you have completed any requested changes.

PRESS

Sincerely, 

Ines

--

Ines Alvarez-Garcia, PhD

Senior Editor

PLOS Biology
